# Uncertainties in projected surface mass balance over the polar ice sheets from dynamically downscaled EC-Earth models

Fredrik Boberg[1], Ruth Mottram[1], Nicolaj Hansen[1,2], Shuting Yang[1], Peter L. Langen[3]

[1]Danish Meteorological Institute, Copenhagen Ø, DK-2100, Denmark
[2]National Space Institute, Kongens Lyngby, DK-2800, Denmark
[3]iClimate, Department of Environmental Science, Aarhus University, Roskilde, DK-4000, Denmark

*Correspondence to*: Fredrik Boberg (fbo@dmi.dk)

**Abstract.** The future rates of ice sheet melt in Greenland and Antarctica are an important factor when making estimates of the likely rate of sea level rise. Global climate models that took part in the fifth Coupled Model Intercomparison
Project (CMIP5) have generally been unable to replicate observed rates of ice sheet melt. With the advent of the sixth Coupled Model Intercomparison Project (CMIP6), with a general increase in the equilibrium climate sensitivity, we here compare two versions of the global climate model EC-Earth using the regional climate model HIRHAM5 downscaling EC-Earth for Greenland and Antarctica. One version (v2) of EC-Earth is taken from CMIP5 for the high-emissions Representative Concentration Pathways (RCP8.5) scenario and the other (v3) from CMIP6 for the comparable high-
emissions Shared Socioeconomic Pathways (SSP5-8.5) scenario. For Greenland, we downscale the two versions of EC-Earth for the historical period 1991–2010 and for the scenario period 2081–2100. For Antarctica, the periods are 1971–2000 and 2071–2100, respectively. For the Greenland Ice Sheet, we find that the mean change in temperature is 5.9 °C when downscaling EC-Earth v2 and 6.8 °C when downscaling EC-Earth v3. Corresponding values for Antarctica are 4.1 °C for v2 and 4.8 °C for v3. The mean change in surface mass balance at the end of the century under these high emissions scenarios is
found to be -290 Gt yr$^{-1}$ (v2) and -1640 Gt yr$^{-1}$ (v3) for Greenland and 420 Gt yr$^{-1}$ (v2) and 80 Gt yr$^{-1}$ (v3) for Antarctica. These distinct differences in temperature change and particularly surface mass balance change are a result of the higher equilibrium climate sensitivity in EC-Earth v3 (4.3 K) compared with 3.3 K in EC-Earth v2 and the differences in greenhouse gas concentrations between the RCP8.5 and the SSP5-8.5 scenarios.

## 1 Introduction

The melt of ice sheets and glaciers now accounts for a greater proportion of observed sea level rise than thermal expansion (Chen et al. 2013, IPCC 2019). With around 150 million people living within 1 meter of current global mean sea level (Anthoff et al. 2006), understanding the likely rate of sea level rise is crucial for planning infrastructure and coastal development. Global climate models (GCMs) that took part in the fifth Coupled Model Intercomparison Project (CMIP5, Taylor et al. 2012) have generally been unable to replicate observed rates of ice sheet melt in Greenland at the present day
(Fettweis et al. 2013) and estimates of sea level contributions from both large polar ice sheets are tracking the upper end of

the range of estimates from these models (Slater et al. 2020). Natural climate variability in the Southern Ocean makes estimating Antarctic surface mass balance (SMB) using climate models complicated and can mask trends related to global warming (Mottram et al. 2021). These uncertainties in current ice sheet response from observations and models give rise to the possibility that the rate of sea level rise over the course of the 21$^{st}$ century may be underestimated in current climate

assessments driven by CMIP5 and earlier model intercomparisons (Slater et al. 2020).

While the CMIP5 experiments were driven by the Representative Concentration Pathways (RCPs, van Vuuren et al. 2011), models in the sixth intercomparison project (CMIP6, Eyring et al. 2016) use a new set of emission and land use scenarios based on socio-economic developments, Shared Socioeconomic Pathways (SSPs, Riahi et al. 2017, O'Neill et al. 2016). Here we use only one of the SSPs called SSP5-8.5, characterized by fossil-fueled development that is the only SSP

consistent with emissions high enough to realize an anthropogenic radiative forcing of 8.5 W m$^{-2}$ in 2100. The total forcing of SSP5-8.5 at 2100 therefore matches that of the RCP8.5 used in CMIP5, but the pathway is different as is the composition in terms of different contributions. For instance, in SSP5-8.5, $CO_2$ emissions and concentrations are somewhat higher than in RCP8.5, but this is compensated for by other constituents such as $CH_4$ and $N_2O$. In this study, we compare results forced by two versions of the EC-Earth coupled global model for RCP8.5 with EC-Earth v2 and SSP5-8.5 with EC-Earth v3. These

two scenarios were chosen as they are the most similar to each other between the CMIP5 and CMIP6 experiments that have been carried out with both model versions.

Several different participating models in the latest generation of GCMs run for CMIP6 (Eyring et al. 2016) have demonstrated an increase in the equilibrium climate sensitivity (ECS) of the models compared to the previous versions in CMIP5 (Voosen 2019, Zelinka et al. 2020). ECS is defined as the time averaged near-surface air warming in response to

doubling $CO_2$ in the atmosphere relative to pre-industrial climate, after the climate system has come into equilibrium. ECS is a commonly used metric to quantify the global warming to increases in atmospheric $CO_2$ including fast feedbacks in the climate system. The higher the ECS, the greater the likelihood of the climate system reaching higher levels of global warming, the smaller the permissible carbon emissions in order to meet a particular climate target. Therefore the ECS is also highly relevant for climate policy.

EC-Earth v3 has a higher ECS of 4.3K compared to 3.3K of EC-Earth v2 from CMIP5 due mainly to a more advanced treatment of aerosols (Wyser et al. 2020b). In this paper, we compare downscaled climate simulations from both versions for Greenland and Antarctica, run with the HIRHAM5 regional climate model (RCM) to examine the impact of the higher ECS on estimates of ice sheet surface mass budget for both Greenland and Antarctica over the 21$^{st}$ century. Higher ECS leads to more rapid atmospheric warming for a given forcing and thus enhanced rates of ice sheet melt. However, as precipitation

often increases in lockstep with a warmer atmosphere, this enhanced melt may be offset to some degree by enhanced snowfall.

The relative performance of EC-Earth on a regional scale in the polar regions has been investigated in several studies, notably by Barthel et al. (2020) for CMIP5 models and also in a new work in preparation by Cecile Agosta (pers. comm) for EC-Earth v3 in the context of the full CMIP6 ensemble. Barthel et al. (2020) shows that EC-Earth v2 has a large bias for

Greenland but with a projected RCP8.5 warming close to the CMIP5 ensemble mean. For Antarctica, Barthel et al. (2020) shows that EC-Earth v2 is among the best models in the atmosphere but performs poorly in ocean subsurface and surface conditions. EC-Earth v2 has also been used in a number of studies with a focus on Greenland and the Arctic, showing that it has an Arctic cold bias. In EC-Earth v3, this Arctic cold bias has more or less disappeared and the current study aims at investigating how this would affect the SMB for Greenland.

The SMB, sometimes also called climatic mass balance, of ice sheets and glaciers is the balance between precipitation, evaporation, sublimation and runoff of snow and glacier ice (Lenaerts et al. 2019). SMB controls the dynamical evolution of ice sheets by driving ice sheet flow from areas of high accumulation to regions of high ice loss. Surface melt and runoff accounts for around 50% of the ice lost from Greenland (Shepherd et al. 2019). When considering the Antarctic Ice Sheet as a whole, dynamical ice loss by calving and the submarine melting of ice shelves are the main mechanisms of ice loss while

SMB processes over the continent, with some exceptions, especially in the Antarctic Peninsula, lead to mass gain. It is important to note that calving and submarine melting of ice shelves do not directly lead to sea level rise as the ice has already left the grounded part of the ice sheet and is floating. However, these ice shelves play an important role in buttressing grounded ice and their loss could trigger large scale retreat and acceleration of marine terminating glaciers. As one mechanism of ice shelf collapse is the accumulation of surface melt leading to hydrofracture as for instance shown in the

collapse of Larsen B (Skvarca et al. 2004), it is important to also calculate SMB over ice shelves, particularly given recent work (Kittel et al. 2021) suggesting large uncertainty over ice shelf SMB in future projections.

As suggested by Fettweis et al. (2013), SMB in Greenland, derived by dynamical downscaling of ERA-Interim reanalysis (Dee et al. 2011) with regional climate models, has a larger runoff component compared with CMIP5 models. This has been attributed to, for instance, a cooler than observed Arctic in EC-Earth v2 by Mottram et al. (2017) or inadequate

representation of Greenland blocking and the North Atlantic Oscillation (NAO) by Hanna et al. (2018). Hofer et al. (2017) and Ruan et al. (2019) also show that cloud properties in climate models are the means by which the NAO modulates ice sheet melt and inadequacies in their representation may be a further source of uncertainty within projections of ice sheet SMB in both Greenland and Antarctica.

Relatively few RCMs have been run or studied in depth for the SMB of Antarctica and results used in international ice sheet

modelling intercomparisons have by and large focused on using results from MAR and RACMO (e.g. Lenaerts et al. 2016, Agosta et al., 2013, 2019; Kittel et al., 2018; Van Wessem et al., 2015, 2018; ). Results of a recent intercomparison of regional models all forced by ERA-Interim (Mottram et al. 2021) show a wide spread of estimates of present day SMB (from 1960 to 2520 Gt yr$^{-1}$) related in large part to different resolutions and precipitation schemes. However, a comparison of

future projections from previous studies (Ligtenberg et al., 2013, Hansen, 2019, Agosta et al., 2013, Kittel et al. 2020)
suggests that on the scale of decades to centuries a clear upward trend in SMB with large interannual and decadal variability is expected due to enhanced snowfall in a warmer climate.

Both the Greenland and the Antarctic ice sheets are important to understand when estimating sea level rise due both to their absolute possible contribution to sea level and for the different timescales and processes that could drive their disintegration. The Antarctic Ice Sheet stores approximately 90% of Earth's freshwater, a potential contribution to mean sea level of 58 m
(Fretwell et al. 2013). Thus, the Antarctic Ice Sheet has the potential to be the single largest contributor to future sea level rise. The Greenland Ice Sheet contains around 7 m of mean sea level rise (Aschwanden et al. 2019) and has in the last two decades seen increasing mass loss (450–500 Gt $yr^{-1}$) due to both large meltwater runoff amounts and enhanced calving from outlet glaciers (Mankoff et al. 2019).

Recent projections from both Greenland and Antarctica have started to include coupled climate and dynamical ice sheet
models from both intermediate complexity models as well as fully coupled regional and global models (Robinson et al. 2012; Vizcaino et al. 2013; Levermann et al. 2020; Le Clec'h et al. 2019; Sloth Madsen et al. 2021). However, most studies still rely on offline ice sheet models forced by higher resolution regional climate models that downscale from global models. In Antarctica, as most ice loss is dynamically driven, SMB is primarily used to provide accurate forcing for ice sheet models. Ice Sheet Model Intercomparison Project for CMIP6 (ISMIP6) models (Goelzer et al. 2018) suggest a wide spread in
projections of sea level rise for Greenland from 70 to 130 mm (Goelzer et al. 2020) including both dynamical and SMB contributions calculated from several different GCMs.

In this study we investigate the differences between two different versions of the GCM EC-Earth, using an identical version of the regional climate model HIRHAM5, for the Greenland and Antarctica ice sheets (see Figure 1). The two EC-Earth models are EC-Earth v2.3 and EC-Earth v3.3 (hereafter referred to as EC-Earth2 and EC-Earth3) and are run for CMIP5 and
CMIP6, respectively. The comparison focuses on temporal changes (end of century relative to a reference period) in temperature, precipitation and the surface mass balance.

In Section 2 we introduce the model domains, the two versions of the GCM EC-Earth as well as the regional climate model HIRHAM5. In Section 3 we present, using time slice experiments and for both Greenland and Antarctica, changes in temperature and precipitation using the two versions of EC-Earth, followed by the resulting changes in surface mass balance
for both ice sheets. The paper ends with a discussion in Section 4 and a conclusion in Section 5.

## 2 Methods and Materials

Here we compare regionally downscaled climate simulations for Greenland and Antarctica (see Figure 1 and Table 1) run with two different versions of EC-Earth and an identical version of the HIRHAM5 RCM. The two EC-Earth models, i.e, EC-

Earth2 and EC-Earth3, are run for CMIP5 and CMIP6, respectively. For reasons of computational cost we run four time slice

experiments with HIRHAM5 driven with EC-Earth forcings for each domain. For Greenland, these cover the period 1990-2010 with historical forcing with both versions of EC-Earth and the period 2080-2100 with CMIP5 RCP8.5 for EC-Earth2 and CMIP6 SSP5-8.5 with EC-Earth3. The historical forcing ends in 2005 for CMIP5 and therefore for the last 5 years of the 1990–2010 period we use RCP4.5 scenario forcing. For Antarctica, the time slice experiments cover the period 1970–2000 with historical forcing and the period 2070–2100 with RCP8.5 and SSP5-8.5. The first year in each time slice experiment is

used for spin-up of atmospheric conditions and not included in the analysis. The difference between time periods for the two regions (1991-2010 vs. 1971-2000 and 2081-2100 vs. 2071-2100) given in Table 1 is a result of the two regions being part of two different studies using the EC-Earth2 downscalings. The subsequent EC-Earth3 downscalings were performed for the same time periods as with the EC-Earth2 downscalings to facilitate a direct comparison between EC-Earth versions.

For the four time slice experiments in Greenland we include an offline spin-up routine on the built-in HIRHAM5 subsurface conditions running for 100 years recycling the first spin-up year from each of the HIRHAM5 simulations. The HIRHAM5 model output for the full time slice simulations for Greenland is subsequently put into a stand-alone offline subsurface model (Langen et al. 2017). Spin-ups of more than 100 years are performed on each of these offline time slice simulations. For the four time slice experiments for Antarctica, there is no initial offline spin-up routine on the built-in HIRHAM5 subsurface

conditions available. Instead, we put the HIRHAM5 output into the stand-alone offline subsurface model (Hansen et al. 2021) where we perform 130 year spin-up for the two historical simulations for Antarctica and an additional 50 years of spin-up for the two scenario simulations. The spin-up time for the scenario runs is shorter in the offline subsurface model since we use the historical spin-up condition as a starting point for the scenario spin-up. The outputs from HIRHAM5 (precipitation and evaporation+sublimation) and the subsurface model (runoff) are used to calculate the SMB of the ice

sheets over these periods in order to be able to compare the different forcings. The HIRHAM5 downscaling in combination with the offline subsurface model gives a more realistic representation of the surface energy balance over the ice sheet as well as surface snow properties and firn-pack processes that lead to retention and refreezing of meltwater. The current version of HIRHAM5 does not have drifting snow implemented.

EC-Earth is a GCM evolving from the seasonal forecast system of the ECMWF (Hazeleger et al., 2010) and developed by a large European consortium. EC-Earth2 is the model used to contribute to CMIP5 and is based on the ECMWF integrated forecasting system (IFS) cy31r1, the NEMO version 2 ocean model and the sea ice model LIM2 (Hazeleger et al, 2012). EC-Earth2 is run on a spectral resolution of T159 (equivalent to ~125 km) and 62 vertical levels up to 5 hPa for the atmosphere, and a 1° x 1° tripolar grid with 46 vertical levels for the ocean and sea ice. The new generation of the EC-Earth model is a

full Earth System model and has been developed to perform CMIP6 experiments. A detailed description of this model is given by Döscher et al. (2020). However, the CMIP6 historical and SSP5-8.5 experiments used in the downscaling in this study were performed with only the GCM configuration i.e, EC-Earth3. EC-Earth3 has upgraded all components of EC-

Earth2, with the IFS cy36r4 for the atmosphere model, the NEMO version 3.6 for the ocean with the sea ice model LIM3 embedded. The EC-Earth3 also runs at a higher resolution than the EC-Earth2. The spatial resolution of the atmosphere is about 80 km horizontally (T255) and 91 vertical levels up to 0.01 hPa for the atmosphere. The ocean model uses the same 1° x 1° tripolar grid as the EC-Earth2 but with 75 vertical levels. The EC-Earth contributed to CMIP5 and CMIP6 historical and scenario experiments with ensembles of 15 and 25 members in total, performed on various platforms by respective consortium members. The differences among these members are only on the initial states which are taken from different snapshots in a 500 year long control run under the pre-industrial condition (Taylor et al, 2012; Eyring et al, 2016). The simulations used in this study were the members r3i1p1 for the CMIP5 and r5i1p1f1 for the CMIP6, carried out at the Danish Meteorological Institute. Figures 2a and 2b show the 1991–2010 mean temperature relative to ERA-Interim for EC-Earth2 and EC-Earth3, respectively. The negative bias over Greenland for EC-Earth2 in Figure 2a is not present for EC-Earth3 in Figure2b. EC-Earth3 has, however a positive bias over Antarctica. Figures 2c and 2d show the difference in the change in 2m temperature and sea surface temperature, respectively, between the EC-Earth3 using SSP5-8.5 and the EC-Earth2 using RCP8.5 at the end of the century relative to the reference period. For 2m temperature in Figure 2c we see a positive difference for both Greenland and Antarctica: between 1 and 3°C along the coastal regions for Greenland and about 1°C in the central parts of Antarctica. There is also a clear difference in sea surface temperature change between the two versions of EC-Earth in Figure 2d: between 1 and 3°C along the coast of Greenland and between 1 and 2°C along the coast of Antarctica. Besides leading to a thinning and a retreat of the ice sheets (if the increase in melt and subsequent runoff outpace the increase in precipitation), these differences in both atmospheric temperature and sea surface temperature are reflected in differences in end-of-winter sea ice extent shown in Figure 3.

Figure 4 shows how EC-Earth2 and EC-Earth3 relates to other CMIP5 and CMIP6 models for changes in temperature and relative changes in precipitation over the ice sheets. We have used one realization for each available GCM containing both a historical run and an RCP8.5/SSP585 scenario run giving a total of 41 CMIP5 model runs including 2 EC-Earth realizations and 28 CMIP6 model runs including 7 EC-Earth realizations. Furthermore, all models are regridded to a common grid and due to the coarse horizontal resolution of the GCMs all land grid points for Greenland and Antarctica are treated as ice sheet points. For the Greenland Ice Sheet the EC-Earth2 model (panel a) is located at the upper part of the scatter plot with the largest changes in precipitation and temperature. This is also true for the EC-Earth3 model (panel c) even though EC-Earth3 gives the lowest changes compared with the other EC-Earth members. For the Antarctic Ice Sheet the EC-Earth2 model (panel b) is located in the middle of the distribution for both precipitation and temperature. This holds also for the EC-Earth3 model (panel d). Comparing Figures 4a and 4c for Greenland and Figures 4b and 4d for Antarctica we see a shift in the ensemble mean temperature going from CMIP5 to CMIP6 (0.3°C for Greenland and 0.4°C for Antarctica) of similar order as when going from EC-Earth2 to EC-Earth3 (0.5°C for Greenland and 0.4°C for Antarctica). We also note that the spread of the EC-Earth members for a specific domain and a specific generation is relatively small compared to the full distribution indicating that sampling issues associated with the relatively short time slices are of minor concern.

The HIRHAM5 regional climate model (Christensen et al. 2006) is based on the HIRLAM7 weather forecasting model (Undén et al. 2002) where the physical routines have been replaced by those within the ECHAM5 climate model (Roeckner et al. 2003). HIRHAM5 uses 31 atmospheric levels and for the Greenland domain, the model is run at a resolution of 0.05° (about 5.5 km) with 20 year long time slices while the Antarctica simulation is run at a resolution of 0.11° (about 12.5 km) with 30 year long time slices. The HIRHAM5 model has previously been validated against observations for Greenland (e.g. Boberg et al. 2018; Langen et al. 2017; Lucas-Picher et al. 2012) and Antarctica (Mottram et al. 2021, Hansen 2019). Boberg et al. (2018) showed that monthly means of observed temperature on the West Greenland ice sheet compare well with the EC-Earth2 downscaling using HIRHAM5 for the period 1993-2010 with a mean bias between +1 and -2°C. Langen et al. (2017) compared 1041 SMB observations from 351 locations in the ablation area of the Greenland Ice Sheet with an ERA-Interim driven HIRHAM5 simulation and found a regression slope of 0.95, a correlation coefficients of 0.75, a RMSE of 0.98 m w.e and a mean bias of -3%, indicating only slightly underestimated net surface mass loss rate. Moreover, comparing to 68 ice cores in the accumulation area of the Greenland Ice Sheet, they found the simulated mean annual accumulation rate to have a -5% bias, 25% RMSE and a correlation coefficient of 0.9. Mottram et al. (2021) showed, using station observations, that ERA-Interim forced HIRHAM5 simulations have a negative bias of -2°C for Antarctica. Using SMB observations, Mottram et al. (2021) found a model mean bias of -20 kg m$^{-2}$ yr$^{-1}$, a RMSE of 101 kg m$^{-2}$ yr$^{-1}$ and a correlation coefficient of 0.81, indicating a small underestimation of the surface mass balance. Mottram et al. (2021) compared Antarctic Ice Sheet SMB estimates taken from five different RCMs forced with ERA-Interim and found that HIRHAM5 had an SMB value for grounded ice about 10% above the ensemble mean and an SMB value for the ice shelves about 4% above the ensemble mean. They concluded that HIRHAM5 SMB values were in the upper range compared with the other models but that the SMB values were almost exactly the same as for the MARv3.10 model, although with a clear difference between its components.

## 3 Results

The temporal and regional changes for temperature, precipitation and SMB taken from dynamical downscalings of EC-Earth2 and EC-Earth3 are presented in this section. As a reference for these variables we use a HIRHAM5 run driven by ERA-Interim, which in turn has been evaluated by Langen et al. (2017) and Mottram et al. (2021) and compared with other similar climate models for Greenland in Fettweis et al. (2020) and Antarctica in Mottram et al. (2021).

### 3.1 Modelled Temperature

Figures 5a and 5c show the annual mean change in 2m temperature for Greenland and Antarctica respectively using HIRHAM5 downscaled with EC-Earth3 for 2081–2100 and 2071–2100 for the SSP5-8.5 scenario relative to the 1991-2010 and 1971-2000 historical runs (cf. Table 2). Figures 5b and 5d show the difference between the changes given in Figures 5a

and 5c and the equivalent change using EC-Earth2 for the same time periods but using the RCP8.5 forcing scenario. Therefore positive values in Figures 5b and 5d do not imply that the scenario period in the EC-Earth3 SSP5-8.5 downscaling is warmer than the scenario period in the EC-Earth2 RCP8.5 downscaling - just that the *change* in temperature is larger from the historical period to the SSP5-8.5 runs compared with the change between the historical simulation and the RCP8.5 runs. The mean change in temperature over the ice sheet is 5.9 °C for Greenland using EC-Earth2 and 6.8 °C using EC-Earth3. For Antarctica the values are 4.1 °C using EC-Earth2 and 4.8 °C using EC-Earth3.

The mean temperature values presented here for the EC-Earth2 and EC-Earth3 downscalings are compared with ERA-Interim downscalings using HIRHAM5 for the reference periods in Table 2. We notice that the temperature for the ERA-Interim driven run is close to the EC-Earth3 driven run for Greenland for the 1991–2010 period. The temperature for the EC-Earth2 downscaling is lower which can be explained by the negative bias in the forcing data. For Antarctica (see Table 2), the downscaled ERA-Interim mean temperature is very close to the downscaled EC-Earth2 mean value while the downscaled EC-Earth3 value is higher due to the positive temperature bias for Antarctica in EC-Earth3. Also note that since ERA-Interim data are only available from 1979 to August 2019, the time period used for the ERA-Interim driven simulation for Antarctica is 8 years shorter than the GCM driven historical runs.

For Greenland (Figure 5b), the change in temperature for the EC-Earth3 run using the SSP5-8.5 scenario is shown to be higher for most of the domain compared with the change in temperature for the EC-Earth2 run using the RCP8.5 scenario. The difference is most pronounced for the northern part of the ice sheet as well as for the non-glacial northern, western and southern coastline. Along the eastern coastline, the difference in temperature change between the two downscalings is close to zero. For Antarctica (Figure 5d), we see similar values as for the Greenland ice sheet except for the eastern part of Antarctica and the western side of the peninsula. This pattern is probably related to the temperature change difference in the GCMs seen in Figure 2c along part of the coastal stretches of Antarctica which in turn could be explained by a change in model bias and/or as a result of aerosol differences between the two GCM versions. As the phase of the southern annular mode (SAM) also controls the spatial variability in precipitation and temperature on annual to decadal scales in Antarctica , the pattern may also reflect different phases of the SAM in the two versions that are, at least in part a result of internal variability rather than climate forcing (Fogt and Marshall, 2020). The largest differences in temperature change for Antarctica are found on the eastern part of the peninsula, the Filchner Ice Shelf and the Ross Ice Shelf.

### 3.2 Modelled Precipitation

For precipitation, we see a positive relative change for both domains (Figure 6a and 6c) using EC-Earth3 and the SSP5-8.5 scenario when downscaling using HIRHAM5 (see Table 2). For Greenland, the largest relative change is found for the northeastern part while the southeastern part of Greenland has changes close to zero. For Antarctica, the largest changes are found in the interior while the coastal areas show a more moderate increase. When comparing the difference in relative changes in precipitation (Figure 6b and 6d) we see negative values for the eastern part of the domains and positive values for

the western parts. These east-west patterns are reminiscent of those in the differences in temperature changes shown in Figure 5b and 5d and in turn are similar to spatial patterns shown in ice core records by Medley and Thomas (2019) which they relate to SAM variability. This suggests that understanding internal variability in global models is important for interpreting SMB projections in Antarctica. For Greenland, the largest positive differences in relative precipitation change are found over the ice sheet in the northwest and to some extent also the southwest and northeast. For Antarctica, the region

with a positive difference in relative precipitation change is more pronounced covering most of the central and western parts.

The precipitation values on grounded ice for the reference periods are compared with downscaled ERA-Interim values using HIRHAM5 in Table 2. For Greenland, the ERA-Interim driven run has a precipitation amount between the two EC-Earth downscalings with EC-Earth2 having a value 7% lower and EC-Earth3 having a value 8% higher than the ERA-Interim

downscaled value. For Antarctica, the EC-Earth2 downscaling has a mean precipitation 11% higher than the ERA-Interim driven run while the downscaled EC-Earth3 has a 33% higher precipitation amount, most likely linked to the positive temperature bias in EC-Earth3 for Antarctica.

### 3.3 Modelled SMB

Figure 7 shows the change in SMB for Greenland (panels a and b) and Antarctica (panels c and d). Figure 7a and 7c shows downscaled EC-Earth2 for the RCP8.5 scenario while Figure 7b and 7d shows downscaled EC-Earth3 for the SSP5-8.5 scenario, all relative to the historical periods (see Table 1). For EC-Earth2 we get a change (2081–2100 relative to 1991–2010) in SMB of -290 Gt yr$^{-1}$ for the entire Greenland Ice Sheet with areas along the western part displaying changes in the range -2 to -1 m yr$^{-1}$. For EC-Earth3 (Figure 7b) almost the entire Greenland Ice Sheet shows a negative change (2081–2100

relative to 1991–2010) in the SMB with values well below -2 m yr$^{-1}$ along the margin. Over the twenty year period at the end of the century for which the model is run, the accumulated SMB anomaly is -1640 Gt yr$^{-1}$. This is equivalent to an additional 4.6 mm of sea level rise per year from the Greenland Ice Sheet at the end of the century, in line with estimates published by Hofer et al. (2020). We also note that the area in the southeast part of the Greenland Ice Sheet with positive contributions for the EC-Earth2 run in Figure 7a is no longer present for the EC-Earth3 run in Figure 7b. For Antarctica on

grounded ice, we get a change (2071–2100 relative to 1971–2000) in SMB of 420 Gt yr$^{-1}$ for the EC-Earth2 simulation (Figure 7c) and a value of 80 Gt yr$^{-1}$ for the EC-Earth3 simulation (Figure 7d). Importantly, the location of the negative SMB in the model coincides with the vulnerable west Antarctic outlet glaciers whose destabilisation could lead to rapid retreat and dynamical ice loss, multiplying many times the effects of the enhanced ice sheet loss.

The SMB values for the reference periods (1991-2010 for Greenland and 1971-2000 for Antarctica) are compared with downscaled ERA-Interim values (1991-2010 for Greenland and 1979-2000 for Antarctica) using HIRHAM5 in Table 2. For Greenland, the ERA-Interim driven run has an SMB value between the two EC-Earth downscalings with EC-Earth2 having a

value 180 Gt yr$^{-1}$ above and EC-Earth3 90 Gt yr$^{-1}$ below the ERA-Interim downscaled value. For Antarctica, the EC-Earth2 downscaling has a mean SMB 220 Gt yr$^{-1}$ above and the EC-Earth3 downscaling 530 Gt yr$^{-1}$ above the ERA-Interim driven

run. The large SMB difference for the EC-Earth3 run for Antarctica is mostly attributable to the difference in precipitation between the ERA-Interim and EC-Earth3 runs but we also note a very high runoff value of 261 Gt yr$^{-1}$ in the EC-Earth3 run.

Also given in Table 2 are the SMB components for the Antarctic ice shelves in parentheses. We see that the two EC-Earth downscalings have comparable numbers for precipitation compared with the ERA-Interim run, which also holds for the EC-

295 Earth2 run for runoff. However, the runoff for the EC-Earth3 run is clearly above the ERA-Interim value owing the warm bias in EC-Earth3. Gilbert and Kittel (2021) used MAR to downscale four GCMs and found, for the historical period, ice shelf SMB values in the range 441 to 526 Gt yr$^{-1}$. Our values using HIRHAM downscaling ERA-Interim and EC-Earth2 are at the lower end of this range while the EC-Earth3 downscaling has a lower SMB value due to the warm Antarctic bias in EC-Earth3. Kittel et al. (2021) presented end of the century changes in the runoff component in the range 32 to 260 Gt yr$^{-1}$

for the grounded ice and 69 to 558 Gt yr$^{-1}$ for the ice shelves. The end of the century changes in the runoff component for our EC-Earth2 downscalings are 242 and 496 Gt yr$^{-1}$ for grounded ice and ice shelves, respectively, placing it near the upper ends of both ranges. The end of century changes in runoff for our EC-Earth3 downscaling for Antarctica are well above these values, with 835 Gt yr$^{-1}$ for grounded ice and 1352 Gt yr$^{-1}$ for the ice shelves. These runoff values are probably a result of the warm bias in EC-Earth3 but partly also inherited from using HIRHAM5 showing high absolute runoff values when

downscaling ERA-Interim (see Table 2).

When looking at yearly sums of the two ice sheet components, precipitation minus sublimation and evaporation and runoff, we can further study the differences between EC-Earth3 and EC-Earth2 for our two model domains (cf. Table 2). For Greenland during the historical period 1991–2010 (Figure 8a), the runoff component for the EC-Earth3 downscaled

simulation is about 400 Gt yr$^{-1}$ larger than for EC-Earth2 while the precipitation minus sublimation and evaporation component has a mean difference of about 120 Gt yr$^{-1}$ with relatively large variations for both simulations. For Greenland during the scenario period 2081–2100 (Figure 8b), the two simulations show a similar difference (now a mean difference of 105 Gt yr$^{-1}$) with respect to the historical period 1991–2010 for the precipitation minus sublimation and evaporation component, whereas runoff shows a steady increase in the difference between the simulations reaching in excess of 2300 Gt

315 yr$^{-1}$ at the end of the century. The downscaled end of century SMB values for EC-Earth2 and EC-Earth3 (196 and -1431 Gt yr$^{-1}$ respectively) can be compared with the ensemble mean SMB, using downscaled CMIP5 and CMIP6 GCMs given by Hofer et al. (2020), of about -300 Gt yr$^{-1}$ for CMIP5 and -1000 Gt yr$^{-1}$ for CMIP6.

For Antarctica during the historical period 1971–2000 (Figure 8c), we see a mean difference of about 460 Gt yr$^{-1}$ for

precipitation minus sublimation and evaporation (cf. Table 2) between the two simulations but for the runoff component, the difference is about 180 Gt yr$^{-1}$ and only small variations are seen, especially for the EC-Earth2 run. For Antarctica during

the scenario period 2071–2100 (Figure 8d), we see that the gap between both precipitation minus sublimation and evaporation as well as runoff increases with time reaching a difference of more than 800 Gt yr$^{-1}$ for both by the end of the century. The downscaled end of century SMB values for EC-Earth2 and EC-Earth3 (2762 and 2730 Gt yr$^{-1}$ respectively) are comparable to the likely SMB range, using CMIP5 and CMIP6 ensembles given by Gorte et al. (2020), of 2630±663 Gt yr$^{-1}$ for CMIP5 and 2418±374 Gt yr$^{-1}$ for CMIP6.

As the large differences between model versions in ΔSMB (1350 Gt yr$^{-1}$ for Greenland and 340 Gt yr$^{-1}$ for Antarctica) are mostly dominated by differences in runoff changes rather than precipitation changes (see Table 2), we attribute them to the warmer reference period for both regions in combination with an approximately 1 °C higher end-of-century warming in both Greenland and Antarctica for EC-Earth3 relative to EC-Earth2. Furthermore, by comparing the spatially averaged temperature values with the runoff values in Table 2, we get an exponential relationship (not shown) that suggests large increases in runoff for relatively small increases in temperature.

## 4 Discussion

Our results show that for two different versions of the driving global model, substantial differences arise in ice sheet surface mass balance at the end of the century when driven by similar greenhouse gas emission pathways. The runoff and precipitation rates at the end of the century over both Greenland and Antarctica are higher, and likely enhanced by the higher temperatures projected under SSP5-8.5 than RCP8.5. The higher temperatures in the EC-Earth3 driven downscalings for the SSP5-8.5 scenario compared with those for the EC-Earth2 driven downscalings for the RCP8.5 scenario are partly caused by a higher equilibrium climate sensitivity (4.3 K compared with 3.3 K in EC-Earth2). The difference between the greenhouse gas emission pathways in SSP5-8.5 and RCP8.5 do also play an important role, however. Gidden et al. (2019) found that the radiative forcing in SSP5-8.5 matched that of RCP8.5 closely but that there were clear differences between the individual greenhouse gas components of the forcing as well as the aerosols. Wyser et al. (2020a) compared an EC-Earth run in CMIP6 (called EC-Earth3 Veg) and the CMIP5 EC-Earth run and concluded that 50% or more of the end of century global temperature increase going from CMIP5 to CMIP6 was due to changes in the greenhouse gas concentrations rather than model changes.

In Figure 4, we compare CMIP5 with CMIP6 ensembles where the EC-Earth members are given as red dots and the two versions used in this study (v2 and v3) have a blue ring around them. Also included are values for the HIRHAM5 downscalings (green dots) for both EC-Earth2 and EC-Earth3 and for both Greenland and Antarctica. By comparing the green dots with the blue rings we see, for Greenland (Figures 4a and 4c), a weakening of the temperature increases (0.8 °C for EC-Earth2 and 0.4 °C for EC-Earth3) after downscaling but at the same time a strengthening of the precipitation increases (8 percentage points for EC-Earth2 and 11 percentage points for EC-Earth3). For Antarctica (Figures 4b and 4d),

however, we see a strengthening of the temperature increases (0.2 °C for EC-Earth2 and 0.5 °C for EC-Earth3) but again a strengthening of the precipitation increases (9 percentage points for EC-Earth2 and 13 percentage points for EC-Earth3). So downscaling leads in all cases to a larger increase in precipitation than what is given in the GCM. For temperature, the warming effect is uniform for both versions of EC-Earth but opposite between Greenland (weakening) and Antarctica (strengthening).

For this study, only one RCM has been used when comparing the downscaling of two GCMs. Future work will expand this to a to a multi-model and multi-member ensemble. However, the HIRHAM5 model has been used for downscaling EC-Earth2 and reanalysis data for both Greenland and Antarctica in a number of studies (Langen et al. 2017, Boberg et al. 2018, Hansen 2019, Mottram et al. 2021) and the model output has been evaluated thoroughly giving it validity for climate modelling as a single member for polar conditions against which other models can be compared.

Our results for Greenland and Antarctica are in line with previous work using MAR (Hofer et al. 2020, Kittel et al. 2020), showing a general increase in melt and runoff rates for Greenland and Antarctica when driven by selected CMIP6 models compared with CMIP5. Hofer et al. (2020) used MAR to downscale six CMIP5 GCMs and five CMIP6 GCMs for the Greenland Ice Sheet  and found an ensemble mean change for the 2081-2100 period of about -700 Gt yr$^{-1}$ (-400 Gt yr$^{-1}$ for the CMIP5 runs and -1100 Gt yr$^{-1}$ for the CMIP6 runs) which is comparable to the values given in Table 2. Kittel et al. (2020) used MAR to downscale two CMIP5 GCMs and two CMIP6 GCMs for the Antarctic Ice Sheet and found that changes in precipitation, runoff and the resulting SMB increase when going from CMIP5 to CMIP6 but with a significant model spread. Table 2 shows similar trends for precipitation and runoff. However, the change in the runoff component for Antarctica (+834 Gt yr$^{-1}$) is clearly higher than  the 32-260 Gt yr$^{-1}$ range given by Kittel et al. (2021), resulting in a negative trend in the SMB change going from EC-Earth2 (+417 Gt yr$^{-1}$) to EC-Earth3 (+80 Gt yr$^{-1}$). However, the scientific argument of the paper is that the change in temperature for the end-of-century high emissions scenario is higher in the EC-Earth3 downscaling compared with the EC-Earth2 downscaling for both Greenland and Antarctica. This difference in temperature change leads to a negative value in the δΔ(SMB) (rightmost column in Table 2). The positive temperature bias for Antarctica in EC-Earth3 is indeed giving very high runoff rates and precipitation amounts but, as seen in Table 2, is not affecting the sign of the δΔ(SMB) value.

Bracegirdle et al. (2015) used 37 CMIP5 models and showed that, due to a large intermodel spread in sea ice area, the change in temperature using the RCP8.5 scenario for Antarctica was in the range 0 to 6 ℃ while the change in precipitation was in the range 0 to almost 40%. This large model spread for future climate change for Antarctica clearly shows the importance of using large model ensembles for climate projections. Analysis of the CMIP6 ensemble for Antarctic sea ice by Roach et al. (2020) showed some improvement in regional sea ice distribution and historical sea ice extent as well as a slight narrowing of the multimodel ensemble spread in CMIP6 compared to CMIP5. Although the wide spread in projections

indicates that a large multi-model ensemble is desirable, comparing two slightly different versions of the same model is helpful to determine which changes may be affected by the difference in the driving models as well as the emissions

pathways, particularly given the difference in ECS between the 2 versions.. The importance of sea surface temperature and sea ice extent to SMB in Antarctica, especially in coastal regions (Kittel et al. 2018) means that variability in ocean and sea ice representation in model projections has large implications for SMB estimates.

## 5 Conclusion

Due to a higher ECS in the driving GCM EC-Earth3 within CMIP6 compared with the driving GCM EC-Earth2 within

CMIP5 together with changes in greenhouse gas concentrations between the RCP8.5 and the SSP5-8.5 scenarios, we find larger changes in both temperature and precipitation for both Greenland and Antarctica in the end-of-century scenario runs compared with the historical simulations. These differences lead to important changes over the polar ice sheets with a change in SMB of around -1640 Gt yr$^{-1}$ for Greenland and +80 Gt yr$^{-1}$ for Antarctica at the end of the century. Comparing these numbers with those obtained from the older EC-Earth2 runs (-290 Gt yr$^{-1}$ for Greenland and +420 Gt yr$^{-1}$ for Antarctica),

suggests that for very high emission pathways, considerable uncertainty still exists for sea level rise contributions from the polar ice sheets due to climate change – even within a single model family. The difference between these two versions corresponds to a sea level rise difference of 3.7 mm per year from Greenland and 1.0 mm per year for Antarctica at the end of the century compared with earlier estimates based on EC-Earth2.

We find that it is difficult to directly compare the downscalings of EC-Earth2 and EC-Earth3 since the forcing conditions are not equal due to revised greenhouse gas concentration scenarios. However this allows us to demonstrate the potentially wide uncertainties on SMB estimates. Moreover the role of natural variability and the impact of climate change on regional circulation patterns that affect SMB are clearly areas that need more research in the future. The results presented here using EC-Earth3 within CMIP6 are therefore important to consider when communicating to the adaptation and mitigation

communities.

## Author contribution

FB, RM, NH and PLL designed the experiments and FB carried them out. FB performed the HIRHAM5 simulations. SY developed the model code for EC-Earth and performed the simulations. FB prepared the manuscript with contributions from all co-authors.

## Acknowledgements

This work is supported by the NordForsk-funded Nordic Centre of Excellence project (award 76654) *Arctic Climate Predictions: Pathways to Resilient, Sustainable Societies (ARCPATH)*. This work has also been supported by the Horizon 2020 EUCP EUropean Climate Prediction system under Grant agreement no. 776613. The work is also supported by the project "Producing RegIoNal ClImate Projections Leading to European Services" (PRINCIPLES, C3S_34b Lot2), part of the Copernicus Climate Change Service (C3S) provided by the European Union's Copernicus Programme and managed by the European Commission. This publication was supported by PROTECT. This project has received funding from the European Union's Horizon 2020 research and innovation programme under grant agreement No 869304, PROTECT contribution number XX. The authors would also like to acknowledge the support of the Danish state through the National Centre for Climate Research (NCKF).

## Data Availability

HIRHAM5 simulation data and SMB model output for all 8 time slice simulations are available upon request to the author. EC-Earth2 and EC-Earth3 data are available on the Earth System Grid Federation portals (eg. https://esg-dn1.nsc.liu.se/projects/esgf-liu/).

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

.

| Domain | Resolution | EC-Earth Forcing | Period |
|---|---|---|---|
| Greenland | 0.05° | v2 historical | 1990–2010 |
| | | v2 RCP8.5 | 2080–2100 |
| | | v3 historical | 1990–2010 |
| | | v3 SSP5-8.5 | 2080–2100 |
| Antarctica | 0.11° | v2 historical | 1970–2000 |
| | | v2 RCP8.5 | 2070–2100 |
| | | v3 historical | 1970–2000 |
| | | v3 SSP5-8.5 | 2070–2100 |

**Table 1: List of all 8 time slice experiments. The Greenland runs are 20 years long while the runs for Antarctica are 30 years long, not counting the first spin-up year in each experiment.**

| Domain | GCM | Period | Temp | Precip | Evap+Subl | Runoff | SMB | ΔSMB | δ(ΔSMB) |
|---|---|---|---|---|---|---|---|---|---|
| Greenland | ERA-Interim | 1991–2010 | -19.3 | 786 | 52 | 435 | 299 | N/A | N/A |
| | EC-Earth2 | 1991–2010 | -23.2 | 728 | 26 | 219 | 482 | -287 | -1350 |
| | | 2081–2100 | -17.3 | 1045 | 32 | 817 | 196 | | |
| | EC-Earth3 | 1991–2010 | -20.2 | 850 | 24 | 620 | 206 | -1637 | |
| | | 2081–2100 | -13.5 | 1125 | 7 | 2549 | -1431 | | |
| Antarctica | ERA-Interim | 1979–2000 | -36.2 | 2356 (632) | 156 (40) | 75 (172) | 2124 (420) | N/A | N/A |
| | EC-Earth2 | 1971–2000 | -35.9 | 2625 (706) | 178 (42) | 79 (210) | 2345 (454) | 417 | -337 |
| | | 2071–2100 | -31.8 | 3395 (881) | 235 (45) | 321 (706) | 2762 (130) | | |
| | EC-Earth3 | 1971–2000 | -32.6 | 3137 (810) | 226 (45) | 261 (593) | 2650 (172) | 80 | |
| | | 2071–2100 | -27.8 | 4111 (1055) | 287 (32) | 1094 (1945) | 2730 (-922) | | |

**Table 2. Temperature (Temp) in °C and SMB components including precipitation (Precip), evaporation+sublimation (Evap+Subl) and surface runoff (Runoff) in Gt yr⁻¹ for grounded ice for all 8 time slice experiments. The temperature is given as a mean for each period while the SMB components are given as mean yearly sums for each period. ΔSMB is the temporal change between the scenario period and the reference period. δ(ΔSMB) is the model difference in ΔSMB between EC-Earth3 and EC-Earth2. Also included are values for the two ERA-Interim driven HIRHAM5 simulations for Greenland and Antarctica. For Antarctica, SMB component numbers in parentheses denote ice shelf values. Note that the time period used for the ERA-Interim driven simulation for Antarctica is 8 years shorter than the GCM driven historical runs.**

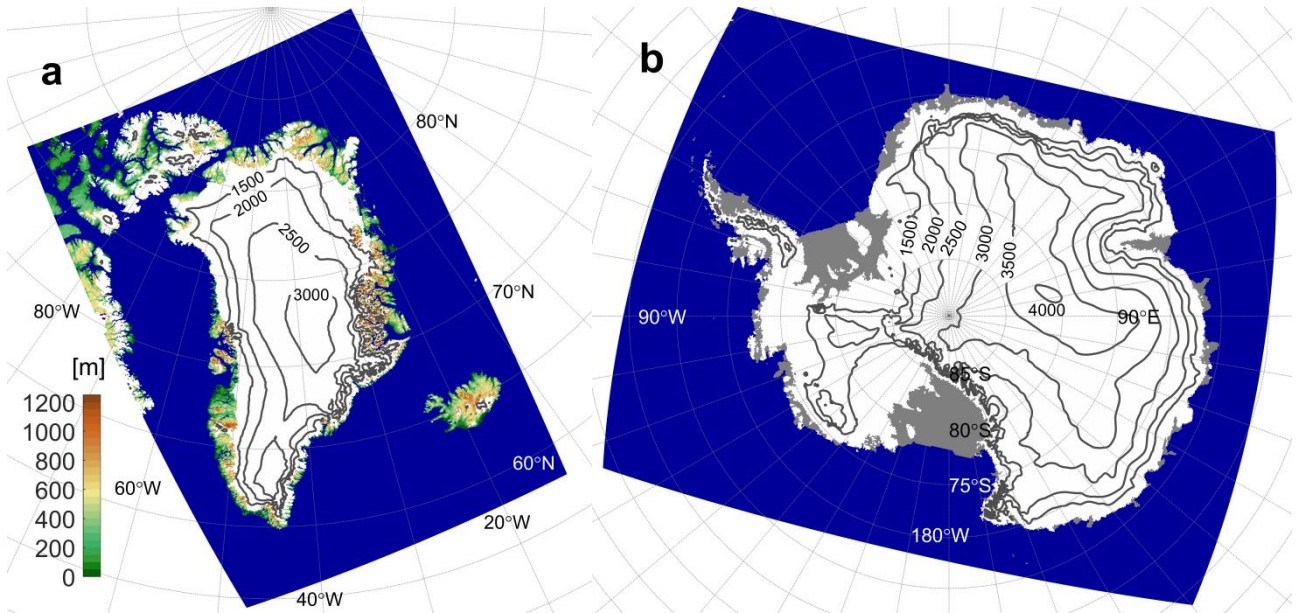

**Figure 1. Topography for the two model domains. Sea points are given in blue, non-glacial land grid points are given in green and brown, Antarctic ice shelves are given in grey while glacial points are given in white with surface elevation contour lines added. The Greenland domain (a) has a model resolution of about 5.5 km (0.05°) while the Antarctica domain (b) has a model resolution of about 12.5 km (0.11°).**

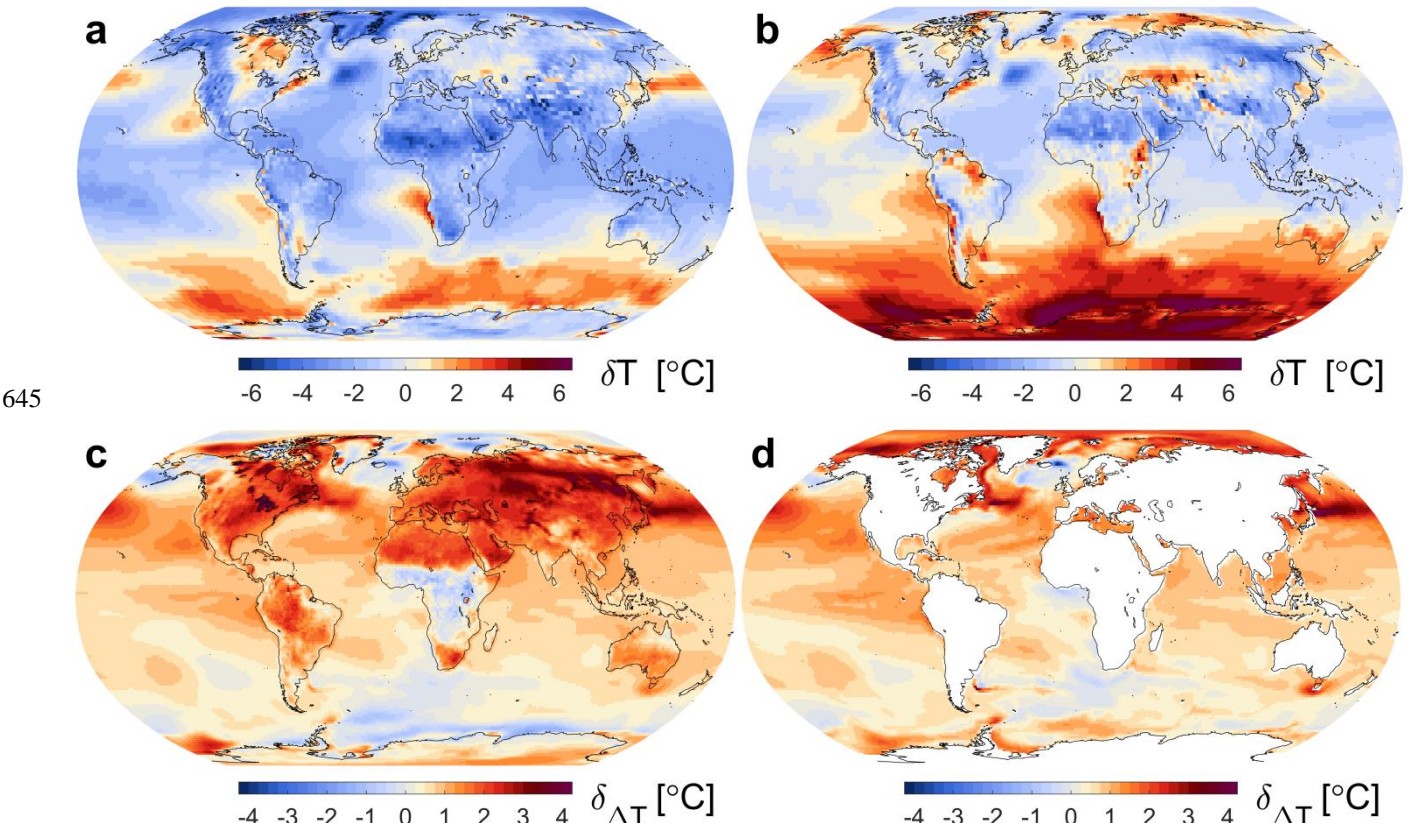

**Figure 2. Temperature bias relative to ERA-Interim for 1991 to 2010 for EC-Earth2 (a) and EC-Earth3 (b). Difference in the change in 2m temperature (c) and sea surface temperature (d) for EC-Earth3 using SSP5-8.5 relative to EC-Earth2 using RCP8.5 for the 2081–2100 period relative to the 1991–2010 historical period.**

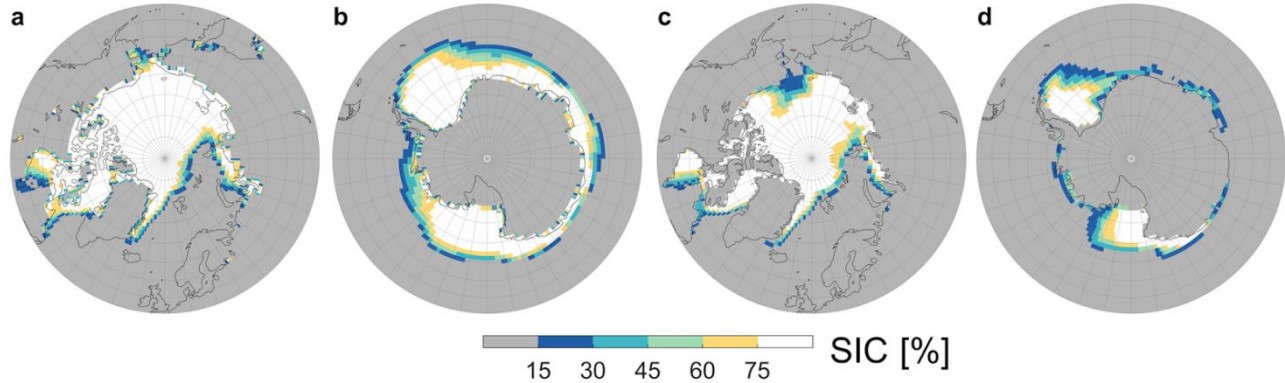

**Figure 3. Mean end-of-winter sea ice extent for the 2081–2100 period. Panels a and b are for the RCP8.5 scenario using Ec-Earth2 and panels c and d are for the SSP5-8.5 scenario using EC-Earth3. Panels a and c are for the month of March while panels b and d are for the month of September.**

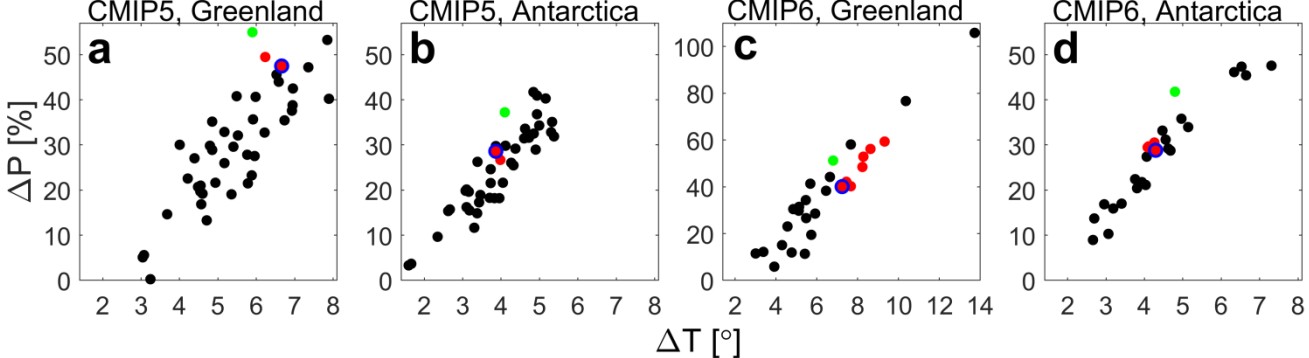

**Figure 4. Relative change in precipitation as a function of change in temperature for 41 CMIP5 and 28 CMIP6 models for the Greenland and Antarctic ice sheets. The change is calculated for the same time periods as for our RCM runs (see Table 1). Panels a and b are for CMIP5 and panels c and d are for CMIP6. Panels a and c are for Greenland while panels b and d are for Antarctica. Red symbols refer to EC-Earth members while all other models are given by black dots. The red and blue dots highlight the EC-Earth members used for downscaling in this study. The green dots are corresponding values for the HIRHAM5 simulations presented in this study.**

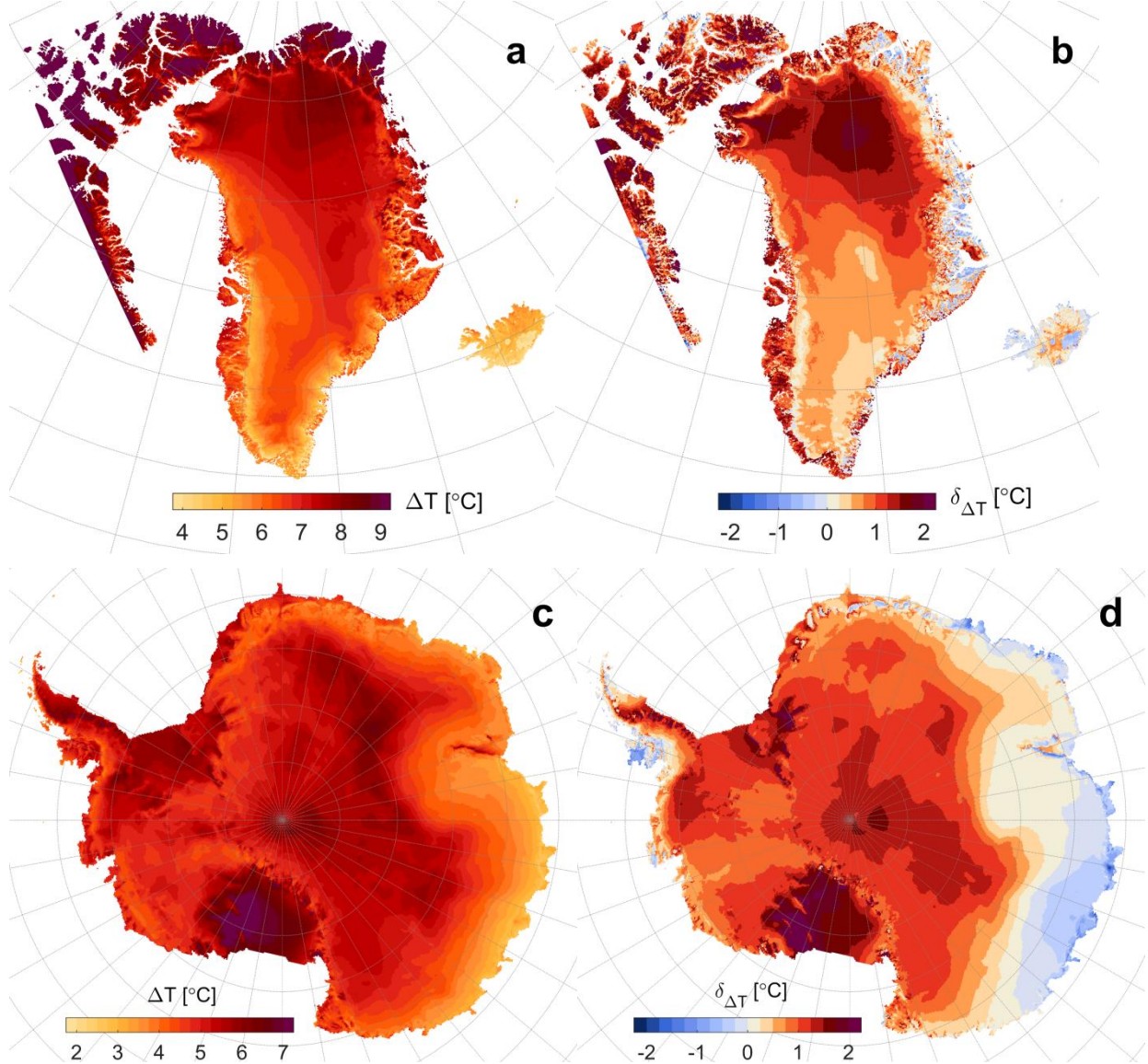

**Figure 5. Change in 2 m temperature for Greenland for 2081−2100 relative to 1991−2010 for the EC-Earth v3 SSP5-8.5 scenario (a). Difference in the change in 2 m temperature for EC-Earth3 SSP5-8.5 relative to EC-Earth2 RCP8.5 (b). Change in 2 m temperature for Antarctica for 2071−2100 relative to 1971−2000 for the SSP5-8.5 scenario (c). Difference in the change in 2 m temperature for SSP5-8.5 relative to RCP8.5 (d). Note that the colorbar limits in panels a and c differ.**

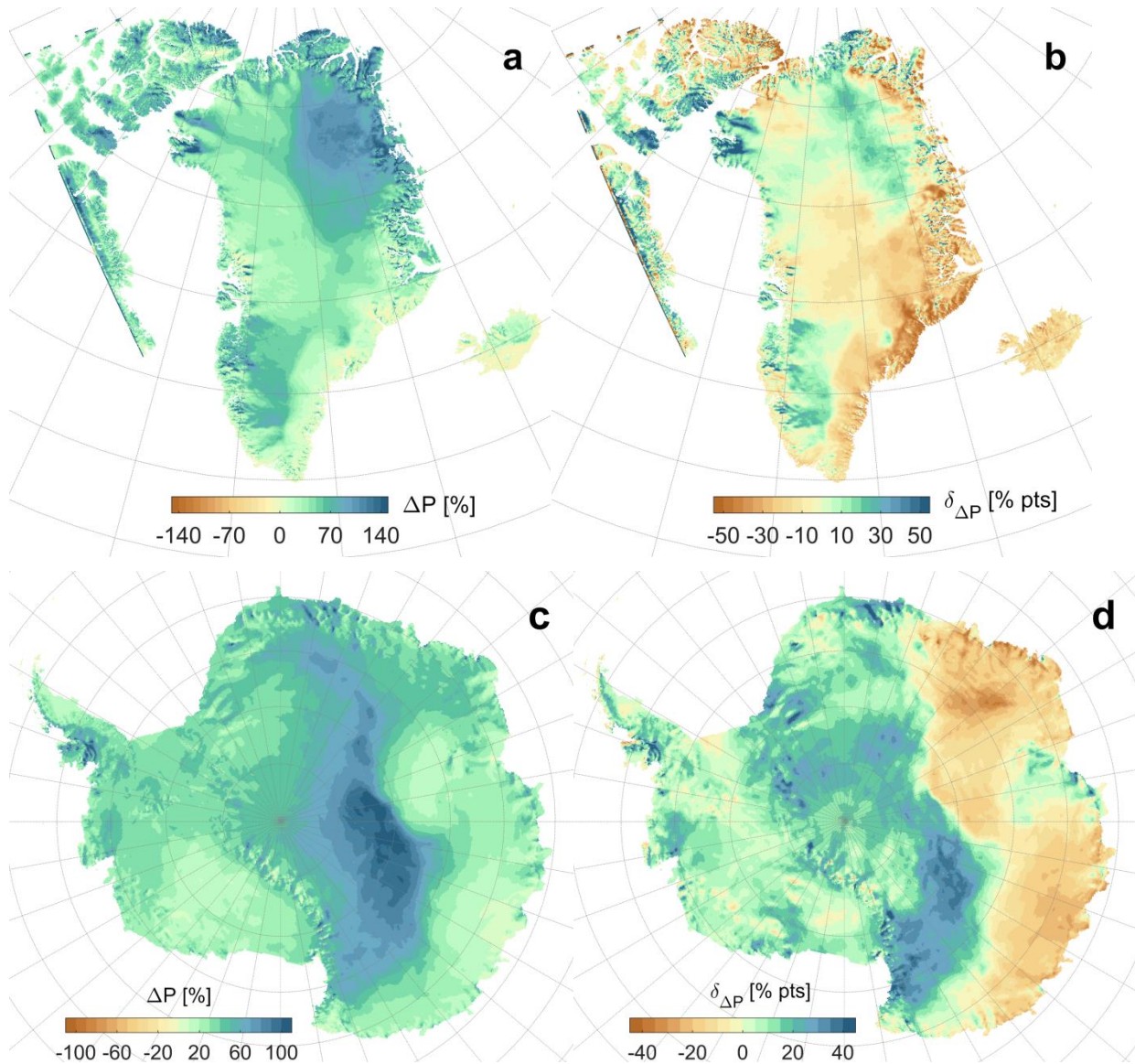

**Figure 6. Relative change in total precipitation for Greenland for 2081−2100 relative to 1991−2010 for the EC-Earth3 SSP5-8.5 scenario downscaling (a). Difference in the relative change in total precipitation for Greenland for the EC-Earth3 SSP5-8.5 relative to EC-Earth2 RCP8.5 downscaling (b). Relative change in total precipitation for Antarctica for 2071−2100 relative to 1971−2000 for the EC-Earth3 SSP5-8.5 scenario downscaling (c). Difference in the relative change in total precipitation for Antarctica for the EC-Earth3 SSP5-8.5 relative to EC-Earth2 RCP8.5 downscaling (d). Note the differences in colorbar limits.**

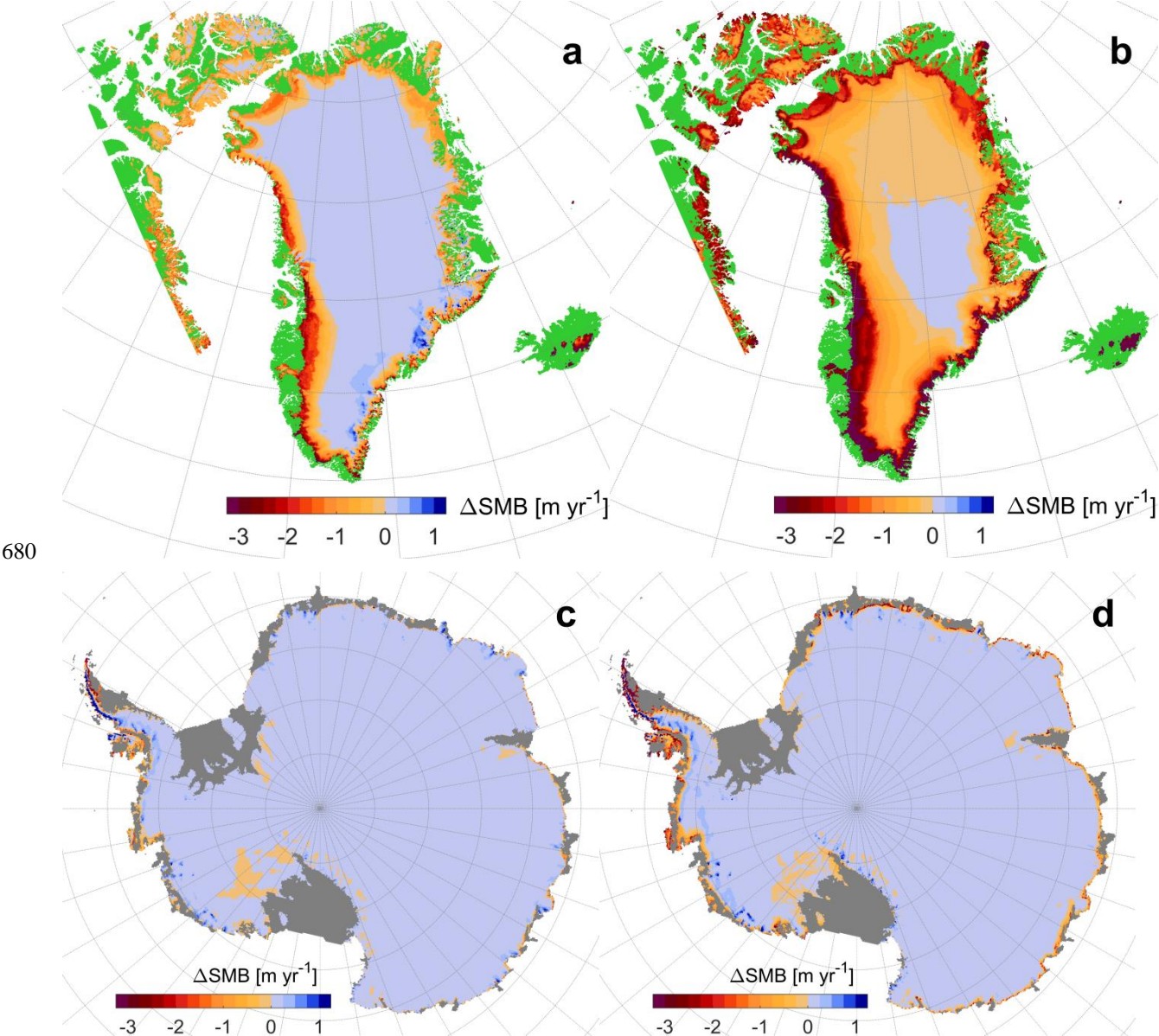

**Figure 7.** Changes in surface mass balance for Greenland for the period 2081−2100 relative to 1991 to 2010 for the EC-Earth2 driven run using RCP8.5 (a) and the EC-Earth3 driven run using SSP5-8.5 (b). Changes in surface mass balance for Antarctica for the period 2071−2100 relative to 1971 to 2000 for the EC-Earth2 driven run using RCP8.5 (c) and the EC-Earth3 driven run using SSP5-8.5 (d). Units are meter water equivalent per year. Green color represents non-glacial land grid points and grey color represents Antarctic ice shelves (cf. Figure 1).

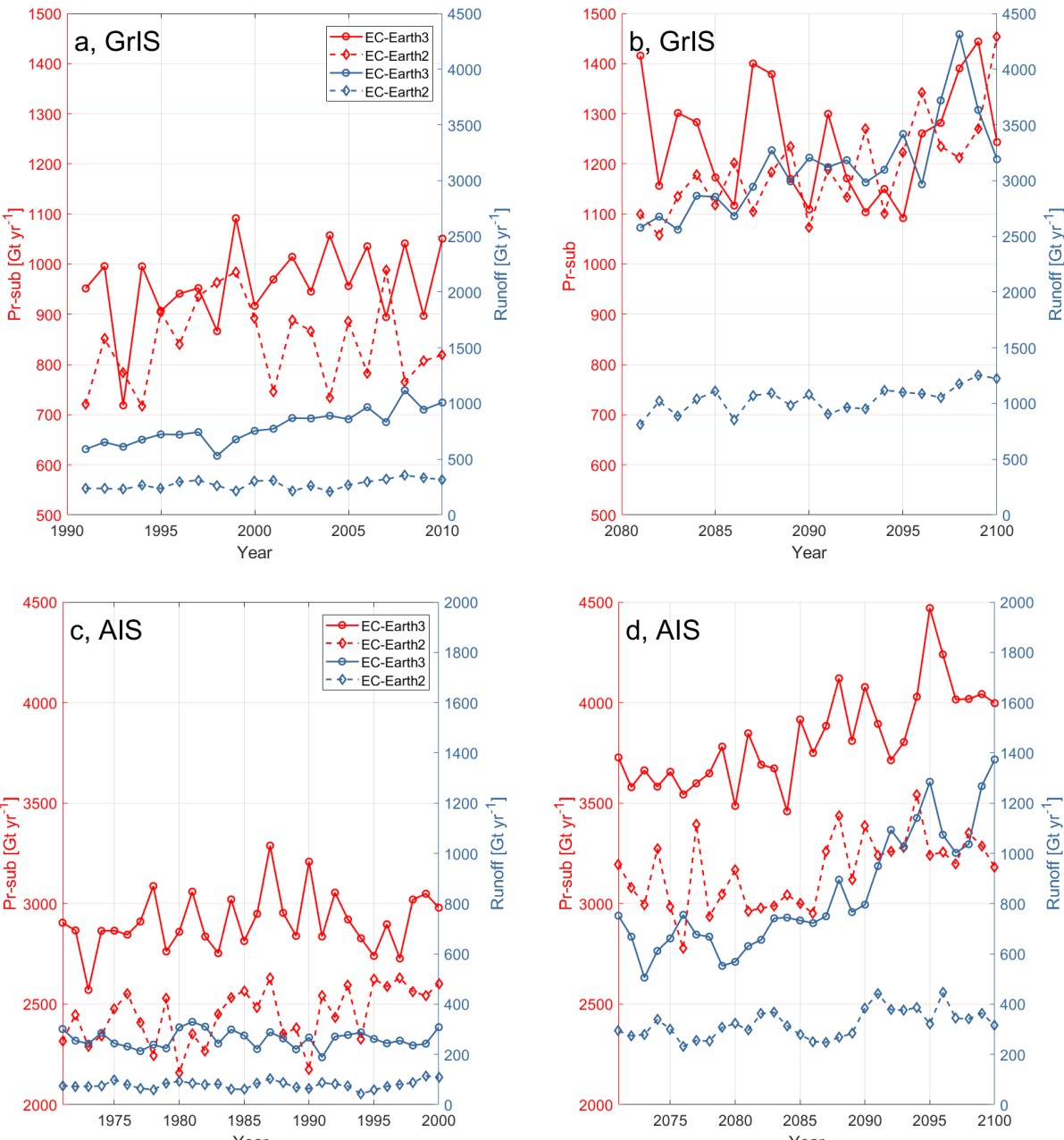

**Figure 8. Integrated values of precipitation minus sublimation and evaporation (in red using the left y-axis) and surface runoff (in blue using the right y-axis) for Greenland (panels a and b) and Antarctica (panels c and d) of HIRHAM5 downscalings of EC-Earth. EC-Earth2 is marked with diamonds and EC-Earth3 is marked with circles.**