# Peer review of "Uncertainties in projected surface mass balance over the polar ice sheets from dynamically downscaled EC-Earth models"

_The Cryosphere, 2021_

## Author Response (AR1)

**Editor**

The anonymous referee #2 states that "Although the regional model results over Greenland for the reference present period seems more reasonable (which is a prerequisite before analyzing projections), those for Antarctica are in significant disagreement with the literature and results for both ice sheets need to be more properly discussed.", which may be an important issue of the discussion (reliability of your model in AIS). I recommend that you attend to this point very carefully. In the last paragraph of Sect. 2, you can explain more in detail about the performance/accuracy of HIRHAM5 forced by ERA-Interim through referring to the paper by Mottram et al. (2021, TC). I think you can say something more about advantages/usefulness of HIRHAM5 applied in AIS. And please tell me and the referee #2 in a convincing manner that HIRHAM5 applied in AIS can be used for analyses of projected future SMB simulated by your model chain.

Reply: We have added ERA-I downscaling comparison from Mottram et al (2021) in the last paragraph of section 2. We do not think that HIRHAM5/SMB values are in significant disagreement with the literature. Yes, the precipitation value for the EC-Earth3 historical downscaling is high but this is due to the warm bias in the forcing GCM and not related to the RCM performance. We have added text comparing our results with literature and hope that this can justify the use of HIRHAM applied in Greenland and Antarctica.

Also, regarding the following comment by anonymous referee #2 "Antarctic ice shelves, over which runoff for the reference present period was exaggerated by several orders of magnitude compared to the existing literature in a previous version of the manuscript, have simply disappeared from the revised version, which is not a satisfying way to deal with this issue.", I recommend that you show some relevant information in a supplementary material, because it is also related to the reliability of your morel. You say that "If needed, we can add another table showing values for the Antarctic ice shelves and compare these values with the literature.", so, please consider including the information in a supplementary material, which can be referred in the last paragraph of Sect. 2.

Reply: The reason why the runoff values are lower in the second version of the manuscript is due to the fact that we have, as a result of a reviewer comment, omitted ice shelves when calculating the total runoff value (and the other SMB components) as well as putting the HIRHAM output through an offline SMB model. So the old values have changed according to a previous reviewer comment. We have included the ice shelf values for the SMB components in table 2 (given in parentheses) for Antarctica and we compare the ice shelf SMB values with the literature in the text.

While revising the manuscript and preparing your rebuttal letter, please always consult your coauthors and find the best solutions.

Reply: Yes, we work closely and are all involved in the review process.

Finally, I have one minor comment on the title:
Please consider adding "dynamically" before "downscaled EC-Earth models". There are some downscaling techniques today, so it is better to specify your technique.

Reply: We agree. We have added the word dynamically.

**Anonymous Referee #1**

General comments:

The authors use HIRHAM5 regional climate model to downscale two EC-Earth models (one for CMIP5 and one for CMIP6), in order to assess projected surface mass balance from Greenland and Antarctica in the future. Whilst this is an important area of research given the uncertainty in future SMB estimates, especially with the difference in CMIP5 and CMIP6 climate sensitivity, there are still very big assumptions being made, and little justification for their research design. After reviewing their previous submission and the changes between versions, there still remains large questions in my mind in terms of the robustness of the results given that only 1 model from each CMIP is used to drafffw conclusions on the future of Greenland and Antarctic SMB. I am pleased to see the evaluation of HIRHAM5 is now included, and have no issue in the choice of RCM (as the authors note, most similar studies use MAR/RACMO, but there is no need for them to be used exclusively). However, I still see no justification for the choice of GCM. There are also a number of other questions remaining on their method design choices and how this influences the results. Therefore, I can't recommend the manuscript for publication in its current state and recommend major revisions.

Reply: We agree that it is a challenge to draw firm conclusions on any differences between CMIP generations using only one GCM. This study is primarily on the difference between two EC-Earth versions but to some extent also on how these EC-Earth versions compare with other CMIP models. When it comes to the choice of GCM, EC-Earth v2 has been used in a number of studies (both as a GCM and as regional downscalings using HIRHAM5) with a focus on Greenland and the Arctic, showing that it has an arctic cold bias (see figure 2). In EC-Earth v3, this arctic cold bias has more or less disappeared (see figure 2) and the current study aims at investigating how this would affect the SMB for Greenland. Unpublished data (Cecile Agosta, pers. comm.) indicates that EC-Earth3 is one of the highest performing GCMs in the CMIP6 ensemble when it comes to replicating observed Arctic climate during the historical period. Given the cold Arctic bias in EC-Earth2, we therefore seek in this publication to understand how the apparently improved version for the Arctic will affect the ice sheet mass budget of Greenland. We also feel that this comparison is worthwhile for the Antarctic ice sheet as many models display this hemispheric asymmetry in performance. It is also not clear that models that perform well compared to observed climate can be said to be more reliable when it comes to future projections as this is more likely to be related to ECS of a given model. Furthermore, we would argue that future climate projections should explore the wide distribution of outcomes from global climate models, rather than focusing on the ensemble mean. As the EC-Earth3 model has a high equilibrium climate sensitivity and the v2 model a rather low one, projections with both model versions likely represent these extremes. Future work will focus on downscaling other GCMs that represent other parts of the CMIP6 model ensemble but here we focus on improvements and changes to one single model as it has evolved from CMIP5 to CMIP6.

Specific major comments:

Methods: Overall, the manuscript would benefit from justification of certain choices you have made in your methods and model setup. I list a number of questions below, which need some justification and also discussion on how your choices may influence your results.

- Model selection: Why EC-Earth, and why the specific realisations that you chose? Whilst I appreciate the high time- and computing power-consumption of downscaling GCM/ESMs with RCMs, there still needs to be some justification of your selections. Efforts are being made to ensure that the 'best' GCM/ESM realisation are chosen for specific regions using selection criteria (see Pickler and Mölg 2021 and earlier references within). Even referencing earlier literature which highlights the success of EC-Earth compared to other models against observations could be cited. What if EC-Earth performs relatively poorly (compared to reanalysis and observation data)

compared with other models? Or what if these specific realisation members (r3i1p1 and r5i1p1f1) are not reflections of the average ensemble for EC-Earth? In the discussion you mention the Southern Annual Mode, but there is no discussion of whether EC-Earth is able to represent the SAM characteristics. In your discussion, you mention the Bracegirdle study which found a large spread in conditions between CMIP5 models (line 315), which further suggests that you need to justify why you have chosen only 1 model. Whilst you do compare EC-Earth to ERA-I for the ice sheets, you don't then compare any other GCMs to ensure that EC-Earth is an appropriate tool. Figure 4 is a step in the right direction, but again doesn't provide any information on whether these two models are the best suggestions of historical/future climate for the ice sheets. Which models are included in Figure 4? Line 154 is quite broad- there are over 600 realisations for CMIP6 models in total, which ones are you using in Figure 4?

Reply: The reply on choice of GCM is given in the reply above. We do not make any attempt in trying to select the best GCM for this study. We want to study EC-Earth and how downscalings of two versions of EC-Earth differ when it comes to ice sheet SMB. As can be seen in figure 4, there are relatively small differences in changes in precipitation and temperature between realization members, so we are convinced that we would get similar results if we had chosen a different member. The models included in figure 4 are one realization for each model available at the time having both historical and RCP85/SSP585 simulations, giving 39 (+ 2 EC-Earth) members for CMIP5 and 21 (+ 7 EC-Earth) members for CMIP6. We will add this to the text and the figure 4 caption.

- Selected time periods: Why are you using different time periods (and different durations) for Greenland and Antarctica? What is the justification for looking at 20 vs 30 years, and why have different time periods in both historical and future runs? If you are trying to compare the SMB and discuss the uncertainties between CMIP5 and CMIP6, why add to the complexity by choosing different time periods? With ERA-I only available from 1979, and therefore this simulation is 8 years shorter than the Antarctica GCM runs (Line 199), why still chose the 1971-2000 period? There is then some discrepancy throughout the manuscript. For instance, Figure 3 shows 2081-2100 but Antarctica uses the period 2070-2100 in other results. Why are there different spin up times for historical and scenario forcings for Antarctica?

Reply: The CMIP5 downscalings for Greenland and Antarctica were part of two separate projects with focus on two different time periods. When we planned for the CMIP6 downscalings we decided to use the same time periods as were used with CMIP5 to save time and computing power. We understand the confusion that can arise from this decision.  When plotting global data (figure 2 and 3) we decided to use the 2081-2100 scenario period since this is common to both regional domains. We did not want to use different time periods in the different panels of figure 3, so we went with 2081-2100 for all 4 panels. Also, figure 2 and 3 were done for 30 year time periods as well and the result is very similar to the shorter periods. When it comes to the spin-up times for Antarctica, the spin-up time for the scenario runs is shorter since we use the historical spin-up condition as a starting point for the scenario runs spin-up. We will add this information to the text.

- Why downscale to such high resolutions when you spend very little time discussing the regional differences? It would be nice to include more about the regional differences between the CMIP runs, as well as just presenting the continent-averaged values.

Reply: Figure 5, 6 and 7 do present regional differences and this is mentioned in the text. But we agree that the differences and similarities can be further discussed. We have added some text in connection with figures 5, 6 and 7 in section 3. We can also mention that there are other ongoing studies looking in more detail on these differences.

- Discussion Ln 310: this paragraph should be more descriptive. To what extent do your results agree with RACMO2 and other models? They all agree with an increase for CMIP6, but are the magnitudes of increase similar in your results to the others? Similarly, in line 298 onwards, you mention the opposite results of other studies, but also do not go into detail about why they disagree. This needs to be discussed more so that the reader can interpret why your results differ or agree.

  Reply: We agree. We have changed the Ln 310 paragraph and added more information here. We have also decided to remove the Ln 298 paragraph since the change in SMB for grounded ice is still positive for the EC-Earth3 downscaling.

Other specific comments:

Throughout: Why v2 and v3? I would recommend abbreviating the model runs to something more intuitive considering CMIP5 and CMIP6 GCMs are used. Perhaps v5/v6 or EC-Earth5 EC-Earth6, so that it is instantly understandable for the reader. It only becomes clear in the third paragraph of the methods that EC-Earth2 and EC-Earth3 are the actual names of the model.

Reply: We mention on line 100 in the introduction section that the EC-Earth2 and EC-Earth3 names come from the model versions EC-Earth v2.3 and v3.3. To be consistent with the model names used in CMIP5 and CMIP6 and to avoid confusion for future versions we would like to stick with 2 and 3.

Ln 93: regional climate models is used rather than an abbreviation here. As there are many abbreviations throughout, perhaps you could avoid using the abbreviation in line 75.

Reply: We have added the RCM and GCM abbreviations in the introduction (lines 28 and 57).

References:

Pickler and Mölg (2021) General circulation model selection technique for downscaling: exemplary application to East Africa. JGR Atmosphere, https://doi.org/10.1029/2020JD033033

**Anonymous Referee #2**

**General comments**

This paper presents projections of the surface mass balance (SMB) of the Greenland and Antarctic ice sheets produced with HIRHAM5 driven by two versions of EC-Earth that took part of the CMIP5 and CMIP6 exercises. Diversifying projections of ice sheet SMB is crucial for a better assessment of the uncertainties related to different members, emission scenarios, or global and regional model biases. While the paper fits well within this scope, in too many places the methodology and analysis still look insufficiently documented to consider publication without major revisions. Considerable improvements compared to the previous version of the manuscript have to be acknowledged for some parts of the paper. However, important comments have also not been taken into account and serious issues persist. Particularly, of major concern are the absence of justification of the chosen GCM and reference periods (in line with RC1), and the incomplete evaluation of the downscaled climate and SMB using HIRHAM5, all already noticed in previous reviewers' reports. Diversity in models is always beneficial, providing that the models perform adequately and are appropriately evaluated. In that sense I encourage the use of regional models other than the widely employed RACMO2 and MAR models, but efforts are still needed in this paper to reach a sufficient level of evaluation for HIRHAM5 products. Although the regional model results over Greenland for the reference present period seems more reasonable (which is a prerequisite before analyzing projections), those for Antarctica are in significant disagreement with the literature and results for both ice sheets need to be more properly discussed. Antarctic ice shelves, over which runoff for the reference present period was exaggerated by several orders of magnitude compared to the existing literature in a previous version of the manuscript, have simply disappeared from the revised version, which is not a satisfying way to deal with this issue. SMB changes in these areas do not contribute directly to SLR but local changes can affect the ice sheet dynamics, especially if these runs are used to drive an ice sheet model. Other minor, though important comments are provided below. I encourage the authors to go through previous reviewers' reports and identify the remaining comments to be considered (they are all meant to help you improving the paper) and adapt the manuscript accordingly or provide a justification otherwise.

Reply: Following a reviewer request from the first round (with which we agreed totally) we changed our analysis to only look at grounded ice and we also added an offline SMB model using HIRHAM5 output as input. As a result, the SMB component values were lowered. We have added numbers for the ice shelves in table 2 and a comparison of the SMB values with the literature. We have also added text comparing HIRHAM5 with other RCMs for the AIS in section 2. Furthermore, we felt that we gave satisfactory answers to all comments from the first round. We are unsure which comments the reviewer is referring to. Can the reviewer provide more details?

**Specific comments**

Section 2: I get that you used EC-Earth because the CMIP runs have been performed at the DMI which is also the affiliation of most of the authors. But this is not enough to justify a choice for a model, and we would want to know more about EC-Earth performance over your regions of interest at least at the RCM boundaries. In the present version we only have a picture of how EC-Earth compares to other CMIP models in terms of relative change in precipitation as a function of change in temperature (which shows that EC-Earth is actually in the upper half of the model ensemble). Here temperature and humidity from the forcing fields are indirectly evaluated through the comparison of average, spatially integrated temperature and precipitation dynamically downscaled by HIRHAM5 against a reference run of HIRHAM5 driven by ERA-I and already evaluated in a previous publication, but what about regional differences ? what about other climatological variables (sea surface conditions, circulation in the forcing itself)?

Reply: The relative performance of EC-Earth on a regional scale in the polar regions has been investigated in several other papers, notably by Barthel et al., 2020 for CMIP5 models and also in a new work in preparation by Cecile Agosta (pers. comm) for EC-Earth v3 in the context of the full CMIP6. EC-Earth v2 has also been used in a number of studies with a focus on Greenland and the Arctic, showing that it has an arctic cold bias (see figure 2). In EC-Earth v3, this arctic cold bias has more or less disappeared (see figure 2) and the current study aims at investigating how this would affect the SMB for Greenland. We are not saying that EC-Earth is the best model to represent the two CMIP generations in any way. We will add more text on this in the introduction section and the references noted above. When it comes to regional differences between the HIRHAM5 downscalings of the two EC-Earth versions, we have added some text linked to figures 5 and 6.

It is not clear how these reference periods have been chosen neither why they differ between Greenland and Antarctica (20 vs 30 years).

Reply: The CMIP5 downscalings for Greenland and Antarctica were part of two separate projects with focus on two different time periods. When we planned for the CMIP6 downscalings we decided to use the same time periods as were used with CMIP5 to save time and computing power. We understand the confusion that can arise from this decision. We will add information on this.

**Minor comments**

P3L63: Instead of distinguishing between evaporation and sublimation, the paper would more gain in exhaustivity in distinguishing between surface sublimation and drifting snow processes, especially for Antarctica as the latters are currently considered to currently drive ablation at the surface of the ice sheet.

Reply: The evaporation and sublimation contributions are given in one single variable. We are not able to separate surface sublimation from other kinds of sublimation. As stated in the manuscript, the current version of HIRHAM5 does not have drifting snow implemented. Also, current studies show little difference on continental scales between models that include drifting snow processes and models that do not.

P4L122: Why is the subsurface model only applied to Antarctica? Please justify. Is it an offline method, i.e., Do the changes in surface properties feed back on the atmosphere ?

Reply: This is a good point. We have now applied the offline model to Greenland as well and the SMB component values are updated in table 2, figure 7, figure 8 and in the text. Yes, the subsurface model is offline and therefore feedback to the atmosphere is excluded.

P4L122-123: I don't really understand the meaning of this sentence as it seems obvious to me that you will use your model results to comment on the SMB change. Do you mean SMB is a diagnostic variable (sum of SMB components vs prognostic = dz/dt * rho) of the model ? Is it computed offline?

Reply: Yes, the sentence is a bit confusing. We get the runoff values from the offline model which is using the HIRHAM output as input. We will make the sentence clearer.

P4L123: "better than" what? Please clarify/justify.

Reply: The sentence will be changed.

P5L143: Why this period and not the historical period?

Reply: The period 1991-2010 is in the historical period for the Greenland runs. This period was chosen for comparison against ERA-Interim in an earlier study when the ERA-I data were only available from 1989 onwards (see Lucas-Picher et al.).

P5L151-152: What are these results telling us regarding the main objective of this paper? what are their implications?

Reply: A sentence on the implications with regards to precipitation and melt has been added.

P5L155, "over the ice sheets": Do you mean each model on their own ice sheet mask ? which region did you use specifically?

Reply: We regrid all model data to a common grid. Due to the poor horizontal resolution of the GCMs we define all land grid points as ice sheet points for Antarctica as well as Greenland. We will add this information to the text.

P5L156: You could comment on the expected consequences for the downscaling.

Reply: We discuss this in the first paragraph on page 10.

P6L164-165: This does not justify why did you choose this specific model though. Note that HIRHAM5 downscalings are on the upper part (or half) for both model ensembles and for both ice sheets.

Reply: You are correct that this does not justify the model choice. See reply on your first specific comment above.

P6L181: This is confusing since the SMB of Antarctica is not negative. Please rephrase.

Reply: The end of the sentence will be rephrased.

P6, Section 3: I would advise being more explicit at the beginning of this section about the approach adopted to evaluate HIRHAM5 runs (HIRHAM5 driven by ERA-I taken as a reference instead of observations etc, performance of HIRHAM5 driven by ERA-I can be found in …). Add and discuss regional differences, instead of only comparing spatially and temporally average values of temperature and SMB components in Table. Compare with other RCMs, discuss the balance between individual components when needed (especially for HIRHAM5 driven by EC-Earth3 in Antarctica: 261 Gt of runoff over the grounded part of the ice sheet vs 3137 Gt of precipitation….)

Reply: We agree. We have added more information in section 3 and in the discussion and given a more detailed description of what is seen in the figures.

P7L198-200: So, does the period over which you compare the temperature between the GCM and the RCM driven by ERA-I differ?

Reply: Yes, for Antarctica we compare 1971-2000 means for HIRHAM5 downscaled using EC-Earth with 1979-2000 means for HIRHAM5 downscaled using ERA-I. However, we get very similar results if we use the 1979-2000 period for both runs. The 1979-2000 sums are between 1 and 1.7% higher than the 1971-2000 values for all three SMB components.

P7L219: Specify that this stands for precipitation on grounded ice only.

Reply: Done.

P8L223-224: Elaborate on the consequences for ice sheet SMB (integrated and regional). What about other surface (ablation) processes relative to the reference run?

Reply: This section deals with precipitation. Please see other sections for other processes (eg. lines 260-265). Also, higher precipitation leads to a higher SMB, which can then lead to a mitigated SLR effect/result from the AIS. Especially, when looking at regional precipitation distributions, because some of the ice shelves are thinning which means that they lose their buttressing effect on the grounded ice. So if the precipitation is falling over the ice shelves it might slow the thinning down.

P8L232: Why is the approach more realistic in a former study? Why it has not been conserved in the present work?

Reply: Thank you for noticing this. All SMB values in the current study are now obtained using this so called "more realistic approach" described in Langen et al. We have modified the text and the values.

P8L238, "420 Gt/yr": Specify that this stands for grounded ice only.

Reply: Done.

P8L242-249: Please discuss the related implications for the present period as well as for projections, show regional differences and add ice shelves to the analysis. Are these differences significative? How do they compare, for instance, to natural variability?

Reply: We have added ice shelf numbers to table 2 and added a paragraph in section 3.3 about comparing ERA-I downscalings for ice shelves with the two EC-Earth downscalings. We have also added text to the discussion section (fourth paragraph).

P9L256-258: How do these numbers compare to previously published projections of GrIS SMB?

Reply: We have added the information to the paragraph.

P9L260-265: Same than previous comment.

Reply: We have added the information to the paragraph.

P10L299-300: see also Kittel et al. (2021) for Antarctica and Fettweis et al. (2013), Hofer et al. (2020) and Noël al. (2021) for Greenland.

Reply: We have removed this paragraph owing that there is no downward trend in SMB using the updated values.

P10L311-313: What about absolute values of the anomalies ? Are they also comparable?

Reply: We have added numbers from Hofer et al (2020) and Kittel et al. (2020) in section 3.3 and in the discussion.

P11L323-325: You give here an argument that undermine your own methodology since you have not discussed the representation of surface oceanic properties in both versions of EC-Earth for the reference period.

Reply: The surface properties are shown in figures 2 and 3 and these have a great influence on the SMB estimates. The warm bias around Antarctica in EC-Earth v3 is clearly affecting the ice sheet conditions. Many CMIP6 models have common issues with clear biases in the Arctic and/or the Antarctic.

P22, Figure 4: Report explanations of letters and colors directly on the plots instead of grouping everything in the caption to improve the readability of the figure.

Reply: Done.

Fettweis, X., Franco, B., Tedesco, M., van Angelen, J. H., Lenaerts, J. T. M., van den Broeke, M. R., and Gallée, H.: Estimating the Greenland ice sheet surface mass balance contribution to future sea level rise using the regional atmospheric climate model MAR, The Cryosphere, 7, 469–489, https://doi.org/10.5194/tc-7-469-2013, 2013.

Hofer, S., Lang, C., Amory, C., Kittel, C., Delhasse, A., Tedstone, A., and Fettweis, X.: Greater Greenland Ice Sheet contribution to globalsea level rise in CMIP6, Nat. Com., 9, 523–528, 2020, https://www.nature.com/articles/s41467-020-20011-8

Kittel, C., Amory, C., Agosta, C., Jourdain, N. C., Hofer, S., Delhasse, A., Doutreloup, S., Huot, P.-V., Lang, C., Fichefet, T., and Fettweis, X.: Diverging future surface mass balance between the Antarctic ice shelves and grounded ice sheet, The Cryosphere, 15, 1215–1236, https://doi.org/10.5194/tc-15-1215-2021, 2021.

Noël, B., van Kampenhout, L., Lenaerts, J. T. M., van de Berg, W. J., & van den Broeke, M. R. (2021). A 21st century warming threshold for sustained Greenland ice sheet mass loss. *Geophysical Research Letters*, 48, e2020GL090471. https://doi.org/10.1029/2020GL090471

---

## Author Response (AR2)

**Editor**

I have received two review reports from the referees and am pleased to tell that the referees are generally satisfied with your responses to the referee's earlier concerns. However, both referees suggest that the manuscript can be much more improved, which I fully agree with.

Anonymous Referee #1 suggests that a paragraph (L. 125 ~ 136), where your subsurface models are introduced, is a bit difficult to follow. The referee raises some important points that should be clarified in your revised manuscript. So, please consider the comments carefully, and revise the manuscript to improve readability.

Anonymous Referee #2 points out that "The high runoff rates for the downscaling of EC-EARTHv3 in Antarctica" suggests there is a (not minor) compensating error in simulating AIS SMB with your model chain. Can you justify that this point does not affect the scientific arguments of this paper?

Please attend to these important issues carefully and revise the paper accordingly. In addition, both referees have provided some useful comments and suggestions to improve the quality/readability of this paper. Please also consider them.

Reply: We agree that the paragraph on lines 125-136 is difficult to follow. We have added text to make it clearer. Also, we have added text to the paragraph dealing with the very high runoff values for the EC-Earth3 downscaling. The scientific argument of the paper is that the change in temperature for the end-of-century high emissions scenario is higher in the EC-Earth3 downscaling compared with the EC-Earth2 downscaling for both Greenland and Antarctica. This difference in temperature change leads to a negative value in the $\delta\Delta$(SMB) (rightmost column in Table 2). The positive temperature bias for Antarctica in EC-Earth3 is indeed giving very high runoff rates and precipitation amounts but, as seen in Table 2, is not affecting the sign of the $\delta\Delta$(SMB) value. We have added this information to the discussion section. The other reviewer comments have also been addressed.

**Anonymous Referee #1**

Overall, the manuscript is much improved over the first and second submissions. The vast majority of reviewer questions/comments have been adequately addressed and the manuscript is now much more thorough and stronger. The authors make it clear why they use only the EC-Earth models from the vast GCM options in CMIP5 and 6. They provide more justification, including an interest in how a cold bias in the CMIP5 version, which was resolved in the CMIP6 version, has an impact on the SMB. Further, they provide more information about why the timings are different between Greenland and Antarctica. They also provide justification for the spin up times, and attempt to clarify the subsurface offline model (although I am still a little confused by this paragraph). Whilst I had no problem with the choice of RCM used, the additional information included to satisfy the other review is a welcome addition to the manuscript too. The authors also provide more regional specifications and have updated their citations based on the time between initial submission and now. I would like to thank the authors for their hard work in each rendition of the manuscript, I hope they also see that it has improved the manuscript and strengthened their work.

I have only very minor suggestions (technical mostly) now and would therefore be happy to recommend it for publication. I address the small issues below:

Minor:
Line 125-136: I am still confused by the subsurface model paragraph. Whilst you attempt to justify why you use it for some of the runs but not others, I am still not sure where the experiment differs and why. I think it may be due to the words 'model output' being used a few times, but I am not sure which model this refers to: HIRHAM, EC-Earth or the Subsurface model. In the previous version, it seemed like you used the subsurface offline model only for Greenland, but now it reads like both locations use it. Is this correct? I think with a re-write, it should likely be solved, so I recommend technical corrections.

Reply: Yes, in the first version we only used the offline subsurface model on the Greenland runs, but then we added a subsurface model to the Antarctic runs in the second version. The main difference in the current version is that an offline spin-up for the built-in HIRHAM5 subsurface us applied for the Greenland runs before the 20 year HIRHAM5 runs are performed; while this is not done for the Antarctic runs since this spin-up routine is not available for the Antarctic domain. The offline stand-alone subsurface model, which is run subsequently using the HIRHAM5 output, includes however a spin-up procedure and is performed on both domains. We have added information to the paragraph.

Technical:
Ln 88: change 'in estimating' to 'when estimating'

Reply: Done

Line 125: Move this first sentence to line 120 (approx.) as I was confused why the dates varied by 1 year in section 2 (line 115-121) until I read this line.

Reply: Done

Ln 197: AIS hasn't been defined before- I would not use an additional abbreviation in this sentence when you already have so many. Similarly, ERA-I was before called ERA-Interim (Ln 190), but from now on you use ERA-I. Make a decision on which to use.

Reply: We have changed to ERA-Interim and replaced AIS with Antarctic Ice Sheet in two places.

Ln 275: perhaps write out the dates here, as I wasn't sure which reference period you relate to, especially when they vary by location.

Reply: We have added the reference periods for the GCM driven runs and the periods used for the ERA-Interim driven runs.

**Anonymous Referee #2**

I'm really glad to read that the authors have addressed most of the reviewers' comments and I would like thank the authors for their efforts. There is still an important, major discussion lacking though on the downscaled SMB components for HIRHAM5 runs over Antarctica (see below), after which the paper would be ready for publication in my opinion. Although the SMB results are indeed in the range of other current estimates, this is not the case for individual SMB components, which suggests that good results are obtained for the wrong reasons. The high runoff rates for the downscaling of EC-EARTHv3 in Antarctica are of particular concern, and may have for instance strong consequences for risk assessment of ice shelves vulnerability. Please expand more on these values. More minor comments are reported below.

L62-64: Please expand briefly in the text on the information brought by these evaluations about the performance of EC-EARTHv2&3 relative to others CMIP models.

Reply: We have added text on the CMIP5 evaluation.

L70-72: This is incorrect. Mass is lost by the ice sheet when the ice flows through the grounding line and becomes afloat. Since ice shelves are already floating terminations of glaciers, occurrence of melting below the ice shelves and/or thinning of ice shelves has an indirect consequence on mass loss through ice sheet dynamics (by triggering the retreat, acceleration and drawdown of marine-terminating glacier) but do not contribute directly to the mass balance in the way suggested here. Please rephrase.

Reply: We have changed the text and added a few sentences here.

L93: I know what you mean here, but strictly speaking this is another shortcut that could be avoided, as melt is not part of the SMB. If all melt refreezes, like in Antarctica, there is no direct link between surface ablation and melt. I suggest to replace with something like "leading to large meltwater runoff amounts" or equivalent.

Reply: We have changed the sentence.

L199-200: This is correct for the SMB, but as a result of a very different balance between its components.

Reply: We have changed the sentence to make this clear.

L283-288: This is where I suggest that your results are in significant disagreement with the literature and here I would call for a more complete comparison with other state-of-the-art estimates of individual SMB components (see Lenaerts et al., 2016; van Wessem et al., 2018; Kittel et al., 2021). The average (1971-2000) ice-sheet integrated runoff values of the EC-EARTv3 downscaling of 854 Gt yr-1 with 593 yr-1 for the ice shelves (Table 2 in the revised manuscript) deserve particular attention. These numbers given for the present climate state are even higher than the average values projected for the end of the century under ssp585 for the warmest ESM in Kittel et al. (2021). More than 3000 Gt yr-1 of projected RU for the AIS by the end of the century is also very high. Moreover, I'm not sure it relates exclusively to a positive temperature bias in EC-EARTH since HIRHAM5 driven by ERA-I already yields high runoff rates (172 Gt yr-1 over the present period, which would correspond to the meltwater fluxes in other RCM estimates, i.e., implying that the whole melt amount runs off entirely). In this reference run, high runoff rates are partly compensated by high snowfall rates in the uppermost range of the whole RCM ensemble (Mottram et al., 2021). Please comment on all of this.

Reply: We have added text to this paragraph highlighting the issues with very high runoff numbers using HIRHAM5 and also downscaling EC-Earth3.

L364-366: The absence of uncertainties associated with your average values complicates a bit the comparison, but as

far as I can read in Kittel et al. (2021) (their Table 1), the highest of the 4 future runoff anomalies amounts to 260 Gt yr-1 for the grounded ice, which differ by much more than 150 Gt yr-1 from the HIRHAM5_EC-EARTH3 future anomaly of 1094-261 = 833 Gt yr-1 reported here. Could you give more details on your calculation?

Reply: We believe there is a misunderstanding here. We are saying that the model mean grounded ice runoff anomaly in the Kittel study is about 150 Gt yr-1 (we simply took the mean of the 4 values) and that this mean anomaly is far less than the runoff from our Ec-Earth3 downscaling. We have changed the sentence to make this clearer. We have also updated our reference of the Kittel study.

Kittel, C., Amory, C., Agosta, C., Jourdain, N. C., Hofer, S., Delhasse, A., Doutreloup, S., Huot, P.-V., Lang, C., Fichefet, T., and Fettweis, X.: Diverging future surface mass balance between the Antarctic ice shelves and grounded ice sheet, The Cryosphere, 15, 1215–1236, https://doi.org/10.5194/tc-15-1215-2021, 2021.

Mottram, R., Hansen, N., Kittel, C., van Wessem, J. M., Agosta, C., Amory, C., Boberg, F., van de Berg, W. J., Fettweis, X., Gossart, A., van Lipzig, N. P. M., van Meijgaard, E., Orr, A., Phillips, T., Webster, S., Simonsen, S. B., and Souverijns, N.: What is the surface mass balance of Antarctica? An intercomparison of regional climate model estimates, The Cryosphere, 15, 3751–3784, https://doi.org/10.5194/tc-15-3751-2021, 2021.

Lenaerts, J.T.M., Vizcaino, M., Fyke, J. et al. Present-day and future Antarctic ice sheet climate and surface mass balance in the Community Earth System Model. Clim Dyn 47, 1367–1381 (2016). https://doi.org/10.1007/s00382-015-2907-4

van Wessem, J. M., van de Berg, W. J., Noël, B. P. Y., van Meijgaard, E., Amory, C., Birnbaum, G., Jakobs, C. L., Krüger, K., Lenaerts, J. T. M., Lhermitte, S., Ligtenberg, S. R. M., Medley, B., Reijmer, C. H., van Tricht, K., Trusel, L. D., van Ulft, L. H., Wouters, B., Wuite, J., and van den Broeke, M. R.: Modelling the climate and surface mass balance of polar ice sheets using RACMO2 – Part 2: Antarctica (1979–2016), The Cryosphere, 12, 1479–1498, https://doi.org/10.5194/tc-12-1479-2018, 2018.